



# CLASSIC v1.0: the open-source community successor to the Canadian Land Surface Scheme (CLASS) and the Canadian Terrestrial Ecosystem Model (CTEM) - Part 2: Global Benchmarking

Christian Seiler[1], Joe R. Melton[1], Vivek K. Arora[2], and Libo Wang[3]

[1]Climate Processes Section, Climate Research Division, Environment and Climate Change Canada, Victoria, BC, Canada
[2]Canadian Centre for Climate Modelling and Analysis, Climate Research Division, Environment and Climate Change Canada, Victoria, BC, Canada
[3]Climate Processes Section, Climate Research Division, Environment and Climate Change Canada, North York, ON, Canada

**Correspondence:** Christian Seiler (christian.seiler@canada.ca)

**Abstract.** The Canadian Land Surface Scheme Including Biogeochemical Cycles (CLASSIC) is an open source community model designed to address research questions that explore the role of the land surface in the global climate system. Here we evaluate how well CLASSIC reproduces the energy, water, and carbon cycle when forced with quasi-observed meteorological data. Model skill scores summarize how well model output agrees with observation-based reference data across multiple

5    statistical metrics. A lack of agreement may be due to deficiencies in the model, its forcing data, and/or reference data. To address uncertainties in the forcing we evaluate an ensemble of CLASSIC runs that is based on three meteorological data sets. To account for observational uncertainty, we compute benchmark skill scores that quantify the level of agreement among independent reference data sets. The benchmark scores demonstrate what score values a model may realistically achieve given the uncertainties in the observations. Our results show that uncertainties associated with the forcing and observations are

10   considerably large. For instance, for 10 out of 19 variables assessed in this study, the sign of the bias changes depending on what forcing and reference data are used. Benchmark scores are much lower than expected, implying large observational uncertainties. Model and benchmark score values are mostly similar, indicating that CLASSIC performs well when considering observational uncertainty. Using the difference between model and benchmark scores as a measure of performance shows that model skill increases in the following order: fractional area burned, runoff, soil heat flux, leaf area index, net shortwave

15   radiation, net ecosystem exchange, above-ground biomass, gross primary productivity, surface albedo, snow water equivalent, net surface radiation, sensible heat flux, net longwave radiation, latent heat flux, and ecosystem respiration. Our results will serve as a baseline for guiding and monitoring future CLASSIC development.





## 1 Introduction

The land surface interacts with the atmosphere through fluxes of momentum, radiation, heat, and mass, the latter including water, trace gases, and aerosols. Land surface models (LSMs) have been developed to simulate these fluxes in global climate models. Early LSMs consisted of simple aerodynamic bulk transfer formulas with prescribed surface parameters that are uniform across continents (Manabe, 1969). Major advancements in LSM development have occurred since then, including the incorporation of ($i$) vegetation effects on the surface energy balance (Dickinson, 1984), ($ii$) vegetation phenology (Sellers et al., 1996), ($iii$) ecological processes (Foley et al., 1996), and ($iv$) nutrient cycles (Goll et al., 2012).

The Canadian Land Surface Scheme Including Biogeochemical Cycles (CLASSIC; v 1.0; Melton et al. 2020) is a state-of-the-art land surface model primarily developed by Environment and Climate Change Canada. CLASSIC is the open-source community model successor to the CLASS-CTEM modeling framework, which consists of the Canadian Land Surface Scheme (CLASS) and the Canadian Terrestrial Ecosystem Model (CTEM). CLASS and CTEM simulate physical and biogeochemical land surface processes, respectively, and together they form the land component of the Canadian Earth System Model (Swart et al., 2019).

Assessing model performance provides invaluable guidance for future model development. Strategies for LSM evaluation have been developed in a number of collaborative research projects, such as the Project for Intercomparison of Land Surface Parameterization Schemes (PILPS) (Henderson-Sellers et al., 1993), the Global Land-Atmosphere Coupling Experiment (GLACE) (Koster et al., 2006), the Protocol for the Analysis of Land Surface Models (PALS), the Land Surface Model Benchmarking Evaluation Project (PLUMBER) (Best et al., 2015), and the International Land Model Benchmarking (ILAMB) project (Collier et al., 2018). The ILAMB approach summarizes model performance across multiple statistical metrics using a dimensionless skill score that ranges from zero to unity, where larger values imply better performance (see section 2 for details).

The ILAMB framework has been adopted by the Global Carbon Project to evaluate the performance of LSMs used for estimating global carbon budgets (Friedlingstein et al., 2019). The project assessed 15 LSMs with respect to gross primary productivity (GPP), ecosystem respiration, net ecosystem exchange (NEE), biomass, soil carbon, leaf area index (LAI), evapotranspiration, and runoff. For CLASS-CTEM, which was one of the 15 models, performance was considerably weaker for LAI and NEE than for other variables. While a low LAI score was also found for most other LSMs, the NEE score was below the multi-model mean value. The variables for which CLASS-CTEM performed best and worst with respect to the multi-model mean were GPP and biomass, respectively. However, skill scores must be interpreted with caution. For instance, low score values may actually reflect uncertainties in model inputs and/or reference data rather than deficiencies in the model. A robust analysis should, therefore, account for uncertainties associated with the forcing and reference data.

A first evaluation of CLASSIC has been presented by Melton et al. (2020), who ran the model for 31 FLUXNET sites using locally observed meteorological data. This present paper provides a comprehensive assessment of model performance on a global scale. Our assessment is based on ILAMB's statistical framework, which we implemented in a new R package referred to as the Automated Model Benchmarking R package (AMBER) (Seiler, 2020). The development of AMBER allowed us to





tailor the ILAMB approach to CLASSIC model output. The resulting benefits include ($i$) a seamless data ingestion that does
not require any pre-processing, ($ii$) the ability to evaluate CLASSIC in different simulation modes (i.e. global runs on a regular
grid, regional runs on a rotated grid, and site-level runs), ($iii$) full control on how the statistical framework is implemented,
and ($iv$) the addition of new functionalities. Our analysis pays special attention to how model performance is affected by
uncertainties in the forcing and reference data.

The remaining paper is organized as follows. Section 2 describes the forcing data, reference data, model structure, and
benchmarking approach. Section 3 assesses model performance for an ensemble of three model runs that differ in their meteo-
rological forcing. For most variables, each ensemble member is compared against more than one observation-based reference
data set. For each variable we assess global, zonal, and seasonal biases, as well as root-mean-square-errors, time lags in sea-
sonal peaks, inter-annual variability, spatial patterns, and corresponding skill scores. Furthermore, we compare model skill

scores against benchmark scores, which are based on reference data only, to assess the extent to which score values are affected
by observational uncertainty. Section 4 elaborates on our principal findings with a particular focus on LAI and NEE, and the
importance of in situ measurements.

## 2   Methods

### 2.1   Canadian Land Surface Scheme Including Biogeochemical Cycles (CLASSIC)

CLASSIC is the open-source community model successor to the CLASS-CTEM modeling framework, which consists of the
Canadian Land Surface Scheme (CLASS) and the Canadian Terrestrial Ecosystem Model (CTEM). CLASS and CTEM are
two fully integrated model components that simulate physical and biogeochemical land surface processes, respectively. A brief
outline of these two components is provided next. CLASSIC's physical component (CLASS) divides the land surface into four
possible subareas, namely bare ground, ground covered by snow, ground covered by canopy, and snow covered by canopy. The

model solves the energy balance for the canopy layer and underlying ground, separately. The canopy consists of a single layer
and is for the results presented here composed of four plant functional types (PFTs), namely needleleaf trees, broadleaf trees,
grasses, and crops. The model output evaluated here is from the offline mode in which we use 20 soil layers ranging from 10
layers of 0.1 m thickness, gradually increasing to a 30 m thick layer for a total ground depth of over 61 m. Water fluxes are
calculated for soil layers within the permeable soil depth of the ground column, but not the underlying bedrock layers, whereas

temperatures are calculated for both soil and bedrock layers. Also calculated are the temperature, mass, albedo, and density of
the single layer snow pack, the temperature and interception and storage of rain and snow on the vegetation canopy, and the
temperature and depth of ponded water on the ground surface.

CLASSIC's biogeochemical component (CTEM) is a dynamic vegetation model coupled to CLASS. CLASS provides
CTEM physical land surface information, including soil moisture, soil temperature and net radiation. CTEM uses this in-

formation to simulate photosynthesis in response to atmospheric $CO_2$ concentration. CTEM considers three live vegetation
components (leaves, stem, and roots) and two dead carbon pools (litter and soil). These five carbon pools are tracked, in the de-
fault CLASSIC configuration used here, for nine PFTs that map directly onto the four PFTs used by CLASS. Needleleaf trees



are divided into their deciduous and evergreen types, broadleaf trees are divided into cold and drought deciduous and evergreen types, and crops and grasses are divided based on their photosynthetic pathways into $C_3$ and $C_4$ versions. The prognostic car-

bon masses of leaf, stem, and root simulated by CTEM are used to calculate the structural vegetation characteristics required by CLASS. The main processes simulated by CTEM include photosynthesis, canopy conductance, tissue turnover, allocation of carbon, and phenology (Arora and Boer, 2005b); dynamic root distribution (Arora and Boer, 2003); maintenance, growth and heterotrophic respiration (Melton et al., 2015); wild fires (Arora and Boer, 2005a; Arora and Melton, 2018); competition for space between PFTs (Arora and Boer, 2006; Melton and Arora, 2016), and land use change (Arora and Boer, 2010). A

more detailed description of the model and its history is documented in Melton et al. (2020). Details on a the computation of radiation fluxes, surface albedo, heat fluxes, LAI, GPP, ecosystem respiration, and biomass are provided in the Appendix.

## 2.2  Meteorological forcing

Meteorological inputs required by CLASSIC include surface downwelling shortwave (SW) and longwave (LW) radiation, near-surface air temperature, precipitation, near-surface specific humidity, surface pressure, and near-surface horizontal wind speed.

Model outputs are potentially sensitive to uncertainties in the meteorological forcing. To account for this uncertainty, we drive CLASSIC with three different quasi-observed meteorological data sets, namely the Climate Research Unit - National Centers for Environmental Prediction version 8 (CRUNCEP; 1901-2016; Viovy 2018), the Climate Research Unit - Japanese 55-year Reanalysis version 2.0 (CRUJRAv2; 1901-2017; Harris et al. 2014; Kobayashi et al. 2015), and the Global Soil Wetness Project Phase 3 (GSWP3) - WFDE5 over land merged with ERA5 over the ocean (W5E5) (GSWP3W5E5; 1901-2016; ISIMIP). All

meteorological forcings are disaggregated from 6-hourly to half-hourly time steps, following the methodology by Melton and Arora (2016).

Zonal mean values suggest that all three forcings are relatively consistent (Figure A1). The corresponding global mean values for CRUJRAv2, GSWP3W5E5, and CRUNCEP are 182, 178, and 183 W m$^{-2}$ for downwelling SW radiation, 304, 309, and 300 W m$^{-2}$ for downwelling LW radiation, 10°C for near-surface air temperature (all three forcings), 59, 64, and 60 mm

per month precipitation, 6.9, 7.4, and 6.9 $\times10^{-3}$ kg kg$^{-1}$ for specific humidity, 947, 948, and 950 hPa surface pressure, and 2.6, 3.3, and 2.6 m s$^{-1}$ horizontal wind speed, respectively.

## 2.3  Simulation protocol

Our simulations are based on the trends in the land carbon cycle project (TRENDY) simulation protocol with time-varying $CO_2$, climate, and land use (S3) (Friedlingstein et al., 2019). The protcol consists of a spin up for the year 1700 and two transient runs

for the periods 1701-1900 and 1901-2016, respectively. The spin up uses a constant atmospheric $CO_2$ concentration of 276.59 ppm, climate data from the early decades of the $20^{th}$ century (i.e. 1901-1920), and a land use map with constant crops and pasture distribution. The first transient run uses the same climate as for the spin up, but time-varying $CO_2$ concentrations and land use for the 1701-1900 period. The second transient run uses time-varying $CO_2$, climate, and land use for the 1901-2017 period.





## 2.4 Reference data

Model outputs are evaluated against in situ observations and globally gridded observation-based reference data (Table 1). The reference database cover 19 variables that form part of the energy, water, and carbon cycle. While most data sets consist of monthly mean values, some are assessed on an annual time scale (streamflow) or present a snap shot in time (vegetation biomass, organic soil carbon mass, and in situ LAI measurements provided by the Oak Ridge National Laboratory). All gridded reference data are spatially interpolated to the spatial resolution of a CLASSIC gridcell ($2.8125° \times 2.8125°$). In situ observations that fall into in the same model grid cell are averaged prior to the comparison against model output. In most cases, the evaluation is based on the grid cell value, which reflects the average conditions across all land cover types located in a grid cell. For above ground biomass, however, a comparison between the grid cell value and forest inventory data is not ideal, as the latter provides biomass estimates for trees only. We, therefore, only consider biomass simulated for trees when evaluating model results against forest inventory data. Details on each reference data set are provided next.

### 2.4.1 In situ observations

The FLUXNET2015 database includes 204 eddy covariance sites with measurements overlapping during the years 1997 and 2014 (Pastorello et al., 2017) (Table 1). The corresponding variables include net surface radiation (RNS), latent heat flux (HFLS), sensible heat flux (HFSS), soil heat flux (HFG) (Figure 1a-d), gross primary productivity (GPP), ecosystem respiration (RECO), and net ecosystem exchange (NEE) (Figure 2 a-c).

Stream flow gauge records are provided by the Global Runoff Data Centre (GRDC) for the world's 50 largest basins (Dai and Trenberth, 2002) (Figure 1e). Measurements were made some time between 1980 and 2010, depending on the basin.

Mortimer et al. (2020) compiled manual gravimetric measurements of snow water equivalent (SNW) collected from 3271 sites located between 42° and 83°N for the 1970-2017 period (Figure 1f).

Above ground biomass (AGB) measurements were provided by the Forest Observation System (FOS) who compiled measurements from 274 permanent research plots (Schepaschenko et al., 2019) (Figure 2d). Measurements are based on allometric relations, and were made between 1999 and 2018, depending on the inventory plot.

Leaf area index (LAI) observations are taken from the Committee on Earth Observation Satellites (CEOS) (Garrigues et al., 2008) and the Oak Ridge National Laboratory (ORNL) (Iio and Ito, 2014) (Figure 2 e and f). The CEOS database consists of 141 sites with measurements during the 1999-2017 period. The values are based on a transfer function that upscales ground LAI measurements to a moderate resolution grid cell using high spatial resolution surface reflectances. The ORNL data set contains a total of 2653 measurements compiled from literature that was published between 1932 and 2012. The values present site-specific maximum ground LAI measurements.

### 2.4.2 Globally gridded reference data

The Moderate Resolution Imaging Spectroradiometer (MODIS) provides globally gridded data sets for white-sky surface albedo (2000-2014), GPP (2000-2016), and LAI (2000-2017). The white-sky albedo (MCD43C3 v006) integrates the Bidirec-





tional Reflectance Distribution Function (BRDF) over all viewing and irradiance directions. The resulting albedo values are properties of the surface that do not depend on the state of the atmosphere (Strahler et al., 1999). Only data with quality flags of less than 3 and solar zenith angles of less than 70° were considered. Evaluating MODIS V6 surface albedo against in situ

measurements shows that biases for the high quality data (full inversion) range from -0.0061 in needleleaf forests to 0.0023 in desserts (Wang et al., 2018).

MODIS-based GPP is provided by Zhang et al. (2017) for the period 2000-2017. The data set estimates GPP as the product of light absorption by chlorophyll and the efficiency that converts the absorbed energy to carbon fixed by plants through photosynthesis. The required inputs include a range of MODIS products (surface temperature, land surface water index, en-

hanced vegetation index, and land cover classification), as well as air temperature and radiation fluxes from NCEP Reanalysis II (Kanamitsu et al., 2002).

MODIS LAI (MOD15A2H, collection 6; Myneni et al. 2015) is based on the inversion of a three dimensional canopy radiative transfer model that simulates surface reflectance from canopy structural characteristics (Knyazikhin et al., 1998). The inversion approach causes LAI to be very sensitive to uncertainties in surface reflectance. Those uncertainties may occur due

to calibration errors, residual atmospheric and cloud contamination, background underneath the canopy, topography, view-illumination geometry effects, and reflectance saturation in dense canopies (Garrigues et al., 2008). MODIS LAI uncertainties of annual mean values range from 23% to 38% across biomes, with a global mean value of 27% for MODIS LAI Collection 5 (Fang et al., 2012).

A second set of globally gridded LAI was provided by Claverie et al. (2016) for the period 1982-2010. This data set is

based on an artificial neural network (ANN) that relates LAI to surface reflectance from the Advanced Very High Resolution Radiometer (AVHRR). The ANN was calibrated with LAI from MODIS (MCD15A2) and in situ LAI measurements from BELMANIP2 (445 sites; Baret et al. 2006). The performance of the algorithm was assessed against in situ observations from the DIRECT database (113; Garrigues et al. 2008).

The Clouds and the Earth's Radiant Energy System (CERES) provides globally gridded SW and LW radiation fluxes at the

surface under all-sky conditions for the period 2000-2012 (Kato et al., 2013). The data set is based on a radiative transfer model that uses satellite-retrieved surface, cloud, and aerosol properties as inputs. The values of these inputs are adjusted within their uncertainty ranges to obtain irradiance values at the top-of-the-atmosphere that are consistent with observations.

The Global Energy and Water Exchanges Surface Radiation Budget (GEWEXSRB) release 3.0 (Stackhouse et al., 2011) provides globally gridded SW and LW radiation fluxes at the surface under all-sky conditions for the period 1984-2007. The

data set is based on a radiative transfer model developed by Pinker and Laszlo (1992), which is driven with satellite observations (cloud parameters and ozone fields) and reanalysis data (Stackhouse et al., 2011). The algorithm scales spectral surface albedos of 12 surface types in such a way that the resulting clear-sky composite broadband top-of-the-atmosphere albedo is consistent with observations from the International Satellite Cloud Climatology Project (Zhang et al., 2019). Surface fluxes have been evaluated against ground-based measurements from sources that include the Baseline Surface Radiation Network and the

Global Energy Balance Archive.





The GOSIF data provided by Li and Xiao (2019) are based on linear correlations between solar-induced chlorophyll fluorescence (SIF) soundings from the Orbiting Carbon Observatory-2 (OCO-2) and GPP measurements from 91 eddy covariance measurements sites from FLUXNET.

The FluxCom data set is an ensemble of globally gridded energy fluxes (HFLS, HFSS, RNS from 2001-2013) and carbon
fluxes (GPP, NEE, and RECO from 1980-2013). Using a range of machine learning methods, FluxCom upscales FLUXNET observations, where remote sensing data and meterological data serve as global predictors. The former includes land surface temperature (LST; MOD11A226), land cover (MCD12Q127), fraction of absorbed photosynthetically active radiation by a canopy (fPAR; MOD15A228), and BRDF-corrected reflectances (MCD43B429) from MODIS. Meterological data employed by FluxCom were taken from the WATCH Forcing Data ERA Interim (WFDEI), Global Soil Wetness Project 3 (GSWP), CRUN-
CEPv837, Global Precipitation Climatology Project (GPCP), and Clouds and the Earth's Radiant Energy System (CERES). The FluxCom values used in the present study are the median values computed over the FluxCom ensembles.

The Conserving Land-Atmosphere Synthesis Suite (CLASSr) provides globally gridded estimates of net surface radiation, latent heat flux, sensible heat flux, soil heat flux, and runoff for the 2003-2009 period (Hobeichi et al., 2019). Each variable presents a weighted mean computed from multiple data products that are, to some extent, observation-based. The data are
observationally constrained with in situ measurements, and each term is adjusted to allow for energy and water balance closure. Net surface radiation, latent heat flux, and sensible heat flux are based on blending data from remote sensing, reanalysis, and land surface models. Soil heat flux estimates are based on reanalysis and land surface models, and runoff is derived from land surface models and hydrological models. The original acronym for the Conserving Land-Atmosphere Synthesis Suite is CLASS. However, in this present paper we use the acronym CLASSr instead to avoid confusion with the Canadian Land
Surface Scheme acronym.

Globally gridded soil moisture estimates provided by Liu et al. (2011) are based on the retrieval characteristics of passive (AMSR-E) and active (ASCAT) microwave satellite data. Remotely sensed estimates are rescaled using the Noah land surface model (Niu et al., 2011) and evaluated with in situ measurements from the International Soil Moisture Network (Dorigo et al., 2011). Values represent conditions of the first 10 cm of the top soil layer.

Gridded estimates of snow water equivalent for the northern hemisphere (NH) are provided by Mudryk (2020). The blended data set is derived from four sources, including a satellite passive microwave radiometer data set (GlobSnow GS3; Takala et al. 2011), reanalysis data (MERRA2; Gelaro et al. 2017), a temperature index model by Brown et al. (2003), and a physical snow pack model referred to as Crocus (Brun et al., 2013). The present study evaluates model output against the blended product, referred to as ECCC, as well as it's four sources.

Global maps of monthly burned area were taken from the the fourth generation of the Global Fire Emissions Database (GFED4; Giglio et al. 2010). The data are based on the 500 m Collection 5.1 MODIS direct broadcast (DB) burned area product (MCD64A1) using the MODIS DB burned-area mapping algorithm by Giglio et al. (2009). The data set has been evaluated against high-resolution Landsat imagery for locations in Southern Africa, Siberia, Central Asia, Alaska, and the western conterminous United States.





A second data set of monthly burnt area is provided by the Fire Disturbance Project of the European Space Agency's Climate Change Initiative programme (Chuvieco et al., 2018). This data set was generated from the MODIS red and near-infrared reflectances (MOD09GQ and MCD14ML, both from collection 6) and thermal anomaly data (MCD14ML). The employed algorithm identifies those grid cells where the presence of wild fires is most evident and then searches for wild fires in close proximity to those grid cells.

CarbonTracker (CT2019) is an inversion model that attempts to reproduce observed atmospheric $CO_2$ concentrations by adjusting $CO_2$ fluxes at the surface. The initial state of terrestrial fluxes are obtained from the Carnegie-Ames Stanford Approach (CASA) biogeochemical model (Potter et al., 1993). Fluxes associated with biomass burning are taken from GFED4S. Observations consists of air samples from 460 sites provided by the GLOBALVIEW+ data product version 5.0 (Masarie et al., 2014). While CT2019 is continuously updated, the period considered in this study ranges from 2000 to 2017.

Globally gridded above-ground biomass estimates are provided by GEOCARBON, which is based on a boreal forest biomass map by Santoro et al. (2015) and two pan-tropical biomass maps from Avitabile et al. (2016) and Saatchi et al. (2011). The GEOCARBON map covers only areas that are dominated by trees in the Global Land Cover 2000 map (GLC2000; Bartholome and Belward 2005). The boreal biomass estimates are based on radar imagery provided by the Envisat Advanced Synthetic Aperture Radar (ASAR). The pan-tropical biomass maps are based on Light Detection and Ranging (LiDAR) observations that were calibrated with in situ measurements of tree allometry. Baccini et al. (2012) upscaled data using a random forest machine learning algorithm and satellite imagery, including the MODIS Nadir BRDF-Adjusted Reflectance (NBAR), MODIS land surface temperature, and shuttle radar topography mission (SRTM) digital elevation data. Saatchi et al. (2011) upscaled in situ measurements using a machine learning algorithm (maximum entropy approach) and satellite imagery, including the MODIS normalized difference vegetation index (NDVI) and leaf area index (LAI) products, global quick scatterometer (QSCAT), and SRTM digital elevation data. Note that we also used the Saatchi et al. (2011) data on its own to evaluate vegetation carbon in the tropics.

        The Harmonized World Soil Database (HWSD) provided by the Food Agriculture Organization (FAO) combines existing regional and national updates of soil information worldwide with the information contained by the FAO Soil Map of the World (Wieder, 2014). The data were processed by Todd-Brown et al. (2013) who computed soil carbon stocks from bulk densities and organic carbon concentrations given in the HWSD for the top 1 m of soil at a $0.5° \times 0.5°$ resolution.

### 2.5   Automated Model Benchmarking R package (AMBER)

The Automated Model Benchmarking R package (AMBER; version 1.0.3) quantifies model performance using a skill score system originally developed by Collier et al. (2018). The method employs five scores that assess the model's bias ($S_{bias}$), root-mean-square-error ($S_{rmse}$), seasonality ($S_{phase}$), inter-annual variability ($S_{iav}$), and spatial distribution ($S_{dist}$). The main steps for computing a score usually include (i) computing a dimensionless statistical metric, (ii) scaling this metric onto a unit interval, and (iii) computing a spatial mean. All scores are dimensionless and range from zero to one, where increasing values imply better performance. These properties allow us to average skill scores across different statistical metrics in order to obtain an overall score for each variable ($S_{overall}$).





Figure 3 provides an overview of our experimental set-up. As a first step we create a small model ensemble by driving
CLASSIC with three different meteorological data sets. The purpose of the ensemble is to determine the sensitivity of model
performance to uncertainties in the meteorological forcing. Next we evaluate each ensemble member using AMBER. The
resulting model scores express how well model and reference data agree. A low score, however, does not necessarily imply
poor model performance, as the lack of agreement may also be due to uncertainties in the forcing data and/or reference data.
To assess to what extent model scores are affected by observational uncertainties, we also run AMBER with reference data
only. The resulting benchmark scores quantify the level of agreement among different reference data sets. Comparing model
scores against benchmark scores then shows how well CLASSIC performs relative to the performance of a reference data set.
The remaining part of this section documents the details of the skill scores employed by AMBER.

### 2.5.1 Bias score ($S_{bias}$)

The bias is defined as the difference between the time-mean values of model and reference data:

$$bias(\lambda, \phi) = \overline{v_{mod}}(\lambda, \phi) - \overline{v_{ref}}(\lambda, \phi), \tag{1}$$

where $\overline{v_{mod}}(\lambda, \phi)$ and $\overline{v_{mod}}(\lambda, \phi)$ are the mean values in time ($t$) of a variable $v$ as a function of longitude $\lambda$ and latitude $\phi$ for
model and reference data, respectively. Nondimensionalization is achieved by dividing the bias by the standard deviation of
the reference data ($\sigma_{ref}$):

$$\varepsilon_{bias}(\lambda, \phi) = |bias(\lambda, \phi)| / \sigma_{ref}(\lambda, \phi). \tag{2}$$

Note that $\varepsilon_{bias}$ is always positive, as it uses the absolute value of the bias. For evaluations against stream flow measurements
the bias is divided by the annual mean rather than the standard deviation of the reference data. This is because we assess
streamflow on an annual rather than monthly basis, implying that the corresponding standard deviation is small. The same
approach is applied to soil carbon and biomass, whose reference data provide a static snap shot in time. In both of these cases,
$\varepsilon_{bias}(\lambda, \phi)$ becomes:

$$\varepsilon_{bias}(\lambda, \phi) = |bias(\lambda, \phi)| / \overline{v_{ref}}(\lambda, \phi). \tag{3}$$

A bias score that scales from zero to one is calculated next:

$$s_{bias}(\lambda, \phi) = e^{-\varepsilon_{bias}(\lambda, \phi)}. \tag{4}$$

While small relative errors yield score values close to one, large relative errors cause score values to approach zero (Figure 4a).
Taking the mean of $s_{bias}$ across all latitudes and longitudes, denoted by a double bar over a variable, leads to the scalar score:

$$S_{bias} = \overline{\overline{s_{bias}}}(\lambda, \phi). \tag{5}$$





### 2.5.2 Root-mean-square-error score ($S_{rmse}$)

While the bias assesses the difference between time-mean values, the root-mean-square-error ($rmse$) is concerned with the residuals of the modeled and observed time series:

$$rmse(\lambda,\phi) = \sqrt{\frac{1}{t_f - t_0} \int_{t_0}^{t_f} (v_{mod}(t,\lambda,\phi) - v_{ref}(t,\lambda,\phi))^2 dt}, \tag{6}$$

where $t_0$ and $t_f$ are the initial and final time step, respectively (Figure A2 a and c). A similar metric is the centralized $rmse$ ($crmse$), which is based on the residuals of the anomalies:

$$crmse(\lambda,\phi) = \sqrt{\frac{1}{t_f - t_0} \int_{t_0}^{t_f} [(v_{mod}(t,\lambda,\phi) - \overline{v_{mod}}(\lambda,\phi)) - (v_{ref}(t,\lambda,\phi) - \overline{v_{ref}}(\lambda,\phi))]^2 dt}. \tag{7}$$

The $crmse$, therefore, assesses residuals that have been bias-corrected (FigureA2 c and e). Since we already assessed the model's bias through $S_{bias}$, it is convenient to assess the residuals using $crmse$ rather than $rmse$. In a similar fashion to the

bias, we then compute a relative error:

$$\varepsilon_{rmse}(\lambda,\phi) = crmse(\lambda,\phi)/\sigma_{ref}(\lambda,\phi), \tag{8}$$

scale this error onto a unit interval (Figure 4a):

$$s_{rmse}(\lambda,\phi) = e^{-\varepsilon_{rmse}(\lambda,\phi)}, \tag{9}$$

and compute the spatial mean:

$$S_{rmse} = \overline{\overline{s_{rmse}}}. \tag{10}$$

### 2.5.3 Phase score ($S_{phase}$)

The skill score $S_{phase}$ assesses how well the model reproduces the seasonality of a variable by computing the time difference ($\theta(\lambda,\phi)$) between modeled and observed maxima of the climatological mean cycle:

$$\theta(\lambda,\phi) = \max(c_{mod}(t,\lambda,\phi)) - \max(c_{ref}(t,\lambda,\phi)), \tag{11}$$

where $c_{mod}$ and $c_{ref}$ are the climatological mean cyle of the model and reference data, respectively (Figure A2b). This time difference is then scaled from zero to one based on the consideration that the maximum possible time difference is six months (Figure 4b):

$$s_{phase}(\lambda,\phi) = \frac{1}{2}\left[1 + \cos\left(\frac{2\pi\theta(\lambda,\phi)}{365}\right)\right]. \tag{12}$$

The spatial mean of $s_{phase}$ then leads to the scalar score:

$$S_{phase} = \overline{\overline{s_{phase}}}. \tag{13}$$





### 2.5.4 Inter-annual variability score ($S_{iav}$)

The skill score $S_{iav}$ quantifies how well the model reproduces patterns of inter-annual variability. This score is based on data where the seasonal cycle ($c_{mod}$ and $c_{ref}$) has been removed (Figure A2 d and f):

$$iav_{mod}(\lambda,\phi) = \sqrt{\frac{1}{t_f - t_0} \int_{t_0}^{t_f} (v_{mod}(t,\lambda,\phi) - c_{mod}(t,\lambda,\phi))^2 dt}, \tag{14}$$


$$iav_{ref}(\lambda,\phi) = \sqrt{\frac{1}{t_f - t_0} \int_{t_0}^{t_f} (v_{ref}(t,\lambda,\phi) - c_{ref}(t,\lambda,\phi))^2 dt}. \tag{15}$$

The relative error, nondimensionalization, and spatial mean are computed next (Figure 4a):

$$\varepsilon_{iav} = |(iav_{mod}(\lambda,\phi) - iav_{ref}(\lambda,\phi))|/iav_{ref}(\lambda,\phi), \tag{16}$$

$$s_{iav}(\lambda,\phi) = e^{-\varepsilon_{iav}(\lambda,\phi)}, \tag{17}$$

$$S_{iav} = \overline{\overline{s_{iav}}}. \tag{18}$$

### 2.5.5 Spatial distribution score ($S_{dist}$)

The spatial distribution score $S_{dist}$ assesses how well the model reproduces the spatial pattern of a variable. The score considers
the correlation coefficient $R$ and the relative standard deviation $\sigma$ between $\overline{v_{mod}}(\lambda,\phi)$ and $\overline{v_{ref}}(\lambda,\phi)$. The score $S_{dist}$ increases from zero to one, the closer $R$ and $\sigma$ approach a value of one (Figure 4c). No spatial integration is required as this calculation yields a single value:

$$S_{dist} = 2(1+R)\left(\sigma + \frac{1}{\sigma}\right)^{-2}, \tag{19}$$

where $\sigma$ is the ratio between the standard deviation of the model and reference data:

$$\sigma = \sigma_{\overline{v_{mod}}}/\sigma_{\overline{v_{ref}}}. \tag{20}$$

### 2.5.6 Overall score ($S_{overall}$)

As a final step, scores are averaged to obtain an overall score:

$$S_{overall} = \frac{S_{bias} + 2S_{rmse} + S_{phase} + S_{iav} + S_{dist}}{1 + 2 + 1 + 1 + 1}. \tag{21}$$

Note that $S_{rmse}$ is weighted by a factor of two, which emphasizes its importance.





## 3 Results

We start our evaluation with a qualitative overview of how the choice of meteorological forcing (CRUJRAv2, GSWP3W5E5, CRUNCEP) and reference data affect global mean biases. Our results show that for 10 out of 19 variables assessed in this study, the sign of the bias changes depending on what forcing and reference data are used (Table 2). Assuming that global values are reasonably accurate if (i) the sign of the bias varies depending on the choice of forcing and/or reference, or (ii) the global mean bias is reasonably small ($\leq 5\%$), we can divide our variables into four groups. The first group includes variables that show accurate global mean values for both gridded and in situ reference data. This includes the variables net surface radiation, sensible heat flux, snow water equivalent, and runoff (Table 2). The second group consists of variables that lack in situ observations, but show accurate values when assessed against gridded reference data. This involves the variables net LW radiation, soil moisture, fractional area burnt, net SW radiation, and vegetation carbon in the tropics. The third group includes variables that show accurate values for either globally gridded data (latent heat flux, GPP, NEE) or in situ reference data (ecosystem respiration, above ground biomass, and LAI). For this group of variables it will be important to assess the differences between evaluations based on gridded and in situ data in more detail. The fourth group includes variables with systematic biases across all evaluations, which includes surface albedo, emissions from wild fires, soil carbon, and soil heat flux. The following sections evaluate biases and other statistical metrics for each variable in detail.

### 3.1 Net surface radiation

Net surface radiation (RNS) biases range from positive to negative values, depending on the choice of forcing (CRUJRAv2, GSWP3W5E5, CRUNCEP) and reference data (CERES, CLASSr, FluxCom, GEWEXSRB, and FLUXNET). For instance, global mean biases range from -15% (-12.07 W m$^{-2}$) when forcing CLASSIC with CRUNCEP and evaluating results against CERES (2000-2012), to +19% (+12.24 W m$^{-2}$) for CRUJRAv2 and CLASSr (2003-2009) (Figure 5). Evaluating model output against in situ measurements by FLUXNET yields mean biases that range from -8% (-6.39 W m$^{-2}$ for CRUNCEP) to +2% (+1.99 W m$^{-2}$ for CRUJRAv2) (Figure 5 and 1a).

Zonal mean values in net surface radiation are in close agreement with the corresponding reference values, with biases that range from -20 W m$^{-2}$ to +20 W m$^{-2}$ (Figure 6). Considering all three model runs and four gridded reference data sets shows that for most grid cells annual mean biases can be both negative and positive, depending on which model run is compared against which reference data (Figure 7a). Hovmoeller diagrams show that CLASSIC captures the observed seasonality well across all latitudes (Figure 8). Monthly biases range from negative to positive values across most latitudes and months, depending on the choice of meteorological forcing and reference data. This suggests that model values are reasonably accurate across space and time when considering the uncertainties associated with the forcing and reference data.

Other statistical metrics confirm that CLASSIC reproduces net surface radiation reasonably well. The relative measure of the centralized root-mean-square error ($\varepsilon_{crmse}$) ranges from 0.30 to 0.49 (unitless), which is lower than for most other variables (Figure 5). Differences in the timing of modeled and observed seasonal peaks ($\theta$) are less than one month. Modeled inter-annual variability is lower compared to CERES, CLASSr, GEWEXSRB, and FLUXNET, and larger than FluxCom (not





shown). The relative inter-annual variability error ($\varepsilon_{iav}$) is reasonably small (0.30 to 0.76, unitless). CLASSIC reproduces the spatial variability of annual mean values well, where the ratio of model and reference standard deviation in space ($\sigma$) is close

to unity (1.00 to 1.19). Also, the spatial correlation coefficients ($R$) range from 0.86 to 0.97. Averaging the resulting scores across all three ensemble members shows that $S_{overall}$ ranges from 0.74 to 0.80 (unitless), depending on the reference data (Figure 9a). The choice of meteorological forcing affects scores by less than 0.1 (Figure 9b). Scores tend to be larger when forcing CLASSIC with GSWP3W5E5 (Figure 9c).

### 3.2 Net shortwave and longwave radiation

Results for net SW radiation (RSS) and LW radiation (RLS) are mainly consistent with findings for net surface radiation described above. Net SW and LW radiation values produced by CLASSIC are reasonably accurate when considering uncertainties associated with meteorological forcing and reference data (CERES and GEWEXSRB). Global mean biases for net SW radiation range from -1% (-1.65 W m$^{-2}$) to -4% (-5.99 W m$^{-2}$) (Figure 5). For net LW radiation, global mean biases range from -17% (-10.42 W m$^{-2}$) to +6% (+4.32 W m$^{-2}$). CLASSIC captures the zonal mean values and seasonal patterns

of both variables well (Figure 6, A4, and A5). However, CLASSIC also shows a systematic negative SW radiation bias in the NH extratropics from December to May (up to -40 W m$^{-2}$) (Figure A4), which is consistent with a surface albedo bias as discussed further below. Results for other statistical metrics and corresponding scores are similar to the ones described for net surface radiation. For net SW radiation, $\varepsilon_{crmse}$ range between 0.22 and 0.38 (unitless), $\theta$ ranges from 0.38 to 0.56 months, $\varepsilon_{iav}$ ranges from 0.11 to 0.53 (unitless), $\sigma$ ranges from 0.94 to 1.13 (unitless), and $R$ ranges from 0.92 to 0.98 (Figure 5). For net

LW radiation, values in $\varepsilon_{crmse}$ and $\theta$ are considerably larger than for net SW radiation, where $\varepsilon_{crmse}$ ranges from 0.51 to 0.65 (unitless) and $\theta$ ranges from 0.63 to 0.76 months. The overall scores are higher for net SW radiation ($S_{overall} \approx 0.80$) than for net LW radiation ($S_{overall} = 0.72$) (Figure 9). The choice of the forcing has a considerable impact on how well the model captures the interannual variability of both, net SW and LW radiation. The difference between the maximum and minimum interannual variability score ($S_{iav}$) ranges from 0.10 to 0.21, depending on the reference data (Figure 9b). For net LW radiation,

the choice of the forcing has also a considerable impact on the bias score ($\Delta S_{bias}$: 0.17-0.2). CLASSIC performs best when forcing the model with GSWP3W5E5 (Figure 9c).

### 3.3 Albedo

Surface albedo (ALBS) shows a positive global mean bias across all three ensemble members and reference data sets (CERES, GEWEXSRB, and MODIS), ranging from +18% to +42% (+0.04 to +0.09, unitless) (Figure 5). This bias occurs mainly in

the NH extratropics, with increasing values towards higher latitudes (+0.2; Figure 6). The bias is most evident during the cold season (+0.4 between 60°N and 80°N from December to January) (Figure A6). Larger biases (+0.5) are present north of 80°N, but this involves only few grid cells. The timing of the bias suggests a possible relation with snow and/or the large solar zenith angle. The performance with respect to other statistical metrics tends to be lower than for net SW radiation. Values for $\varepsilon_{crmse}$ range between 0.98 and 1.12 (unitless), $\theta$ ranges from 1.67 to 2.18 months, $\varepsilon_{iav}$ ranges from 0.38 to 0.48 (unitless), $\sigma$

ranges from 1.12 to 2.04 (unitless), and $R$ ranges from 0.77 to 0.91 (Figure 5). The corresponding overall score $S_{overall}$ ranges





between 0.48 and 0.59, depending on the reference data (Figure 9). The $S_{overall}$ values are not very sensitive to the choice of forcing ($\Delta S_{overall} = 0.01$).

### 3.4 Heat fluxes

CLASSIC reproduces the partitioning of net surface radiation into sensible (HFSS), latent (HFLS), and soil heat flux (HFG)
reasonably well. Global mean values in sensible heat flux simulated by CLASSIC are likely to be reasonable, with biases that range from negative to positive values, depending on the meteorological forcing (-23% to +35%, -8.33 W m$^{-2}$ to +10.82 W m$^{-2}$) (Figure 5 and 10). Driving CLASSIC with CRUNCEP causes biases to be negative for all three reference data sets, while the opposite is true when running the model with CRUJRAv2. For latent heat flux, the sign of the bias is consistent across all ensemble members, with negative biases when compared against FluxCom (-12% to -18%, -5.35 W m$^{-2}$ to -8.32 W m$^{-2}$)
and FLUXNET (-7% to -13%, -3.16 W m$^{-2}$ to -5.83 W m$^{-2}$) and reasonably small positive biases when evaluated against CLASSr (+5% to +11%, +1.55 W m$^{-2}$ to +3.67 W m$^{-2}$). Biases in soil heat flux are negative across all ensemble members and both reference data sets (CLASSr and FLUXNET) (-32% to -90%). However, it must be noted that the corresponding absolute values are small (-0.14 W m$^{-2}$ to -1.31 W m$^{-2}$), implying that the impact of annual mean soil heat flux biases on the surface energy balance is minor. The relative partitioning of net surface radiation into sensible and latent heat flux is well
reproduced, with reasonably small positive and negative biases, depending on the meteorological forcing (Figure 10b). A more detailed assessment of each heat flux is provided next.

CLASSIC reproduces the zonal mean patterns of sensible and latent heat flux well, with zonal mean biases of about $\pm$ 20 W m$^{-2}$ (Figure 11). For sensible heat flux, zonal mean biases range from positive to negative values across all latitudes, depending on the meteorological forcing (Figure 11 and A7a). For latent heat flux, biases are primarily positive when compared against
CLASSr and negative for FluxCom, regardless of the forcing (Figure 11 and A7b). Zonal mean patterns of soil heat flux data from CLASSr are not well captured, with biases that become increasingly negative towards higher latitudes (-4 W m$^{-2}$ in the Arctic) (Figure 11 and A8a).

CLASSIC captures the seasonal cycle of all three heat fluxes reasonably well (Figure A9, A10, and A11). The negative latent heat flux biases are stronger during the warm season in both hemispheres. For soil heat fluxes, CLASSIC tends towards a more
intense seasonal cycle in the extratropics, with positive biases in the warm and negative biases in the cold season.

Values of $\varepsilon_{rmse}$ are similar among sensible and latent heat flux (0.45 to 0.77, unitless) and larger for soil heat flux (0.85 to 0.97) (Figure 5). For all three heat fluxes, time lags in seasonal peaks are close to one month. The inter-annual variability of sensible and latent fluxes tends to be smaller compared to the CLASSr reference data set and larger compared to FluxCom (not shown), implying that model values lie within the uncertainty of the reference data. The corresponding $\varepsilon_{iav}$ values range
from 0.25 to 2.42 (unitless), where the larger values are obtained when evaluating CLASSIC against FluxCom (Figure 5). For soil heat flux, modeled interannual variability is very similar to the corresponding reference values (not shown), leading to low values of $\varepsilon_{iav}$ (0.28 to 0.36). Values of $\sigma$ and $R$ are reasonable for sensible heat flux ($\sigma$: 0.74 to 1.26, $R$: 0.78 to 0.89) and latent heat flux ($\sigma$: 0.94 to 1.08, $R$: 0.84 to 0.97), and poor for soil heat flux ($\sigma$: 0.15 to 0.17, $R$: 0.04 to 0.21). Consistently, the corresponding $S_{overall}$ values are larger for latent and sensible heat flux (0.64 to 0.73) and lower for soil heat flux (0.53)





(Figure 9). From all three heat fluxes, sensible heat flux is most sensitive to the choice of forcing, with $\Delta S_{overall}$ equal to 0.04 when driving the model with GSWP3W5E5.

## 3.5 Soil moisture

Global mean values in soil moisture (MRSLL) of the top 10 cm are very close to the corresponding reference values from ESA (1979-2017), with biases ranging from -1% (-0.28 kg m$^{-2}$, CRUNCEP) to +5% (about +1 kg m$^{-2}$, CRUJRAv2 and

GSWP3W5E5) (Figure 5). Zonal mean values are well reproduced with a tendency towards a positive bias north of 60°N (+10 kg m$^{-2}$) (Figure 11). This bias is caused by clusters of grid cells located in different parts of the Acric (Figure A8b). The positive bias is present in all ensemble members and occurs from June to September (Figure A12). Other statistical metrics confirm that the performance does not differ largely among ensemble members (Figure 9). Values of $\varepsilon_{rmse}$ and time lags of seasonal peaks ($\theta$) are larger compared to most other variables ($\varepsilon_{rmse} \leq 1.5$, $\theta \approx 1.5$ months). The spatial variability is larger

than in the reference data ($\sigma \approx 1.44$, unitless) and correlation coefficients are relatively low ($R \approx 0.62$). The corresponding overall score is 0.52 and is not very sensitive to the choice of forcing ($\Delta S_{overall} = 0.01$) (Figure 9).

## 3.6 Runoff and streamflow

Global mean runoff (MRRO) biases range from +1% (<0.005 kg m$^{-2}$ day$^{-1}$, CRUNCEP) to +34% (0.22 kg m$^{-2}$ day$^{-1}$, GSWP3W5E5) when evaluated against CLASSr (2003-2009) (Figure 5). This difference is consistent with the larger precipita-

tion values found for the GSWP3W5E5 forcing. CLASSIC reproduces the zonal mean pattern of gridded reference runoff from CLASSr well, with mainly positive biases when forcing CLASSIC with GSWP3W5E5 (Figure 11). Smaller, including negative zonal mean biases occur when forcing CLASSIC with CRUNCEP. Consistently, the sign of the bias varies for many grid cells depending on the forcing (Figure A13a). Furthermore, the sign of the bias varies for most latitudes and months (Figure A14), suggesting that modeled runoff is within the uncertainty range of the meteorological forcing. CLASSIC reproduces the

spatial patterns of runoff reasonably well, with $\sigma$ values between 1.04 and 1.14 and $R$ values between 0.87 and 0.88 (Figure 5). The timing of the seasonal peaks, however, differ by more than two months and $\varepsilon_{rmse}$ values exceed one when using CLASSr as reference data. The overall scores averaged over the three ensemble members is 0.53, with a better score when driving CLASSIC with CRUNCEP and worse when using GSWP3W5E5 as a forcing data set (Figure 9).

Summing up the annual runoff from all grid cells located within each of the world's 50 largest river basins and comparing

those values against annual mean stream flow at the corresponding river mouths yields biases that range between -14% (-0.11 kg m$^{-2}$ day$^{-1}$, CRUNCEP, GRDC) to +20% (+0.16 kg m$^{-2}$ day$^{-1}$, GSWP3W5E5, GRDC). For most basins, the sign of the bias remains constant regardless of the forcing (Figure 1e for CRUJRAv2 as an example). CLASSIC reproduces the spatial patterns of streamflow reasonably well, with $\sigma$ values close to 1 and $R$ values ranging between 0.87 and 0.92 (Figure 5). The overall scores averaged over the three ensemble members is 0.69. Performance is better when driving CLASSIC with

CRUNCEP and worse when using GSWP3W5E5 as a forcing data set (Figure 9).



### 3.7 Snow water equivalent

Global mean snow water equivalent (SNW) biases range from -14% (-0.49 cm, CRUNCEP-driven run evaluated against MERRA2) to +70% (+1.88 cm, GSWP3W5E5-driven run evaluated against Crocus) for the 1981-2017 period (Figure 5). Evaluating CLASSIC against in situ measurements compiled by Mortimer et al. (2020) shows that the mean bias can be neg-

ative (-10%, -0.88 cm, CRUNCEP) or positive (+23%, +2.07 cm, GSWP3W5E5), depending on the forcing (Figure 5 and 1f). The larger snow water equivalent values in GSWP3W5E5-driven results are consistent with the larger precipitation values found for this forcing. For most evaluations, CLASSIC reproduces zonal mean values well when driven with CRUJRAv2 or CRUNCEP (Figure 11). The sign of the bias ranges from positive to negative values for most grid cells (Figure A13b). Positive biases tend to be larger in Eastern Canada, Northern Scandinavia and Northern Siberia, and during NH spring (Figure A15b).

Other statistical metrics confirm that CLASSIC's ability to reproduce snow water equivalent tends to be weaker when driven with GSWP3W5E5 ($\varepsilon_{rmse}$: 0.55 to 0.91, $\theta$: 0.38 to 0.77, $\varepsilon_{iav}$: 0.19 to 0.61, $\sigma$: 0.68 to 1.68, and $R$: 0.54 to 0.87) (Figure 5). The corresponding $S_{overall}$ values range from 0.62 to 0.69, with lower scores when driving CLASSIC with GSWP3W5E5 and higher scores when using CRUJRAv2 (Figure 9). Site-level based score values are similar to grid-level score values, with an overall score of 0.66.

### 3.8 Gross primary productivity

Globally accumulated annual mean values of GPP computed from different reference data are of similar magnitude, with 115 Pg C yr$^{-1}$ for FluxCom (1980-2013), 121 Pg C yr$^{-1}$ for MODIS (2000-2016), and 130 Pg C yr$^{-1}$ for GOSIF (2000-2017) (Table 3). The corresponding model biases are reasonably small, with negative values when comparing CLASSIC against GOSIF (-2% to -10%), and positive values when evaluating against MODIS (<+1% to +10%) or FluxCom (+4% to +14%). An assessment

with eddy covariance data shows that biases range from -6% to -10% (Figure 5). CLASSIC captures zonal mean GPP patterns well with no systematic bias pattern among evaluations (Figure 12). Annual mean biases range from negative to positive for most grid cells, depending on the forcing and reference data (Figure A16a). Hovmoeller diagrams show that CLASSIC captures the seasonal GPP cycle well across all latitudes, with no systematic biases (Figure A17). Values in $\varepsilon_{rmse}$ are for all evaluations less than one, and time lags of seasonal peaks are about one month on average (Figure 5). CLASSIC reproduces the inter-annual

variability of GPP well when compared against FLUXNET, GOSIF, and MODIS ($\varepsilon_{iav} < 0.6$). Larger $\varepsilon_{iav}$ values are found when evaluating the model against FluxCom. However, it must be noted that interannual variability patterns from FluxCom are not reliable (Jung et al., 2019b). Spatial patterns of GPP are well captured, with values of $\sigma$ close to one and correlation coefficients around 0.91. The corresponding overall scores range from 0.58 (FluxCom) to 0.67 (MODIS), with larger scores when forcing CLASSIC with CRUJRAv2 (Figure 9).

### 3.9 Ecosystem respiration

Annual mean biases of ecosystem respiration (RECO) are reasonably small across FLUXNET sites, with mean biases ranging from +5% to +12%, depending on the forcing (Figure 5). Looking at the spatial distribution of FLUXNET sites suggests that





the positive biases are most evident in the NH extratropics rather than in the tropics (Figure 2b). Larger biases are present when

comparing CLASSIC with FluxCom, however, those values are not reliable as FluxCom underestimates ecosystem respiration

in the tropics (Jung et al., 2019b). Other statistical metrics derived from FLUXNET data suggest that CLASSIC reproduces

ecosystem respiration reasonably well ($\varepsilon_{rmse} \approx 0.75$, $\theta < 1$, $\varepsilon_{iav} \approx 0.66$, $\sigma \approx 1.0$, $R \approx 0.69$) (Figure 5). The corresponding

overall score equals 0.64 (Figure 9). This score value is insensitive to the choice of forcing.

### 3.10   Net ecosystem exchange

Globally accumulated annual mean NEE computed from CT2019 equals -4.2 Pg C yr$^{-1}$ for the period 2000-2016 (Table 3).

The corresponding value from FluxCom is -24.9 Pg C yr$^{-1}$ for the period 1980-2013, which is unrealistic, and due to an

underestimation of ecosystem respiration in the tropics (Jung et al., 2019b). CLASSIC's estimate of annual mean NEE ranges

from -4.2 to -4.6 Pg C yr$^{-1}$ (2000-2016), which corresponds to a bias of about -1% to -10% when compared against CT2019,

depending on the meteorological forcing. However, much larger biases apply when evaluating CLASSIC against FLUXNET

(+88%; Figure 2c and 5).

Zonal mean NEE patterns resemble values from CT2019 to a certain degree (Figure 12). The strength of the carbon sink is

modeled to be stronger in the tropics (-0.15 g C m$^{-2}$ day$^{-1}$) and weaker in the extratropics (+0.10 g C m$^{-2}$ day$^{-1}$) compared

to CT2019. FLUXNET data confirm the tendency towards a positive bias, i.e. weaker sink, in the NH extratropics (Figure 2c).

In the tropics, however, the sign of the bias is mixed, and the number of sites is too small to confirm whether or not CLASSIC

overestimates the strength of the carbon sink at low latitudes.

Hovmoeller diagrams suggest that CLASSIC reproduces the general pattern of the seasonal NEE cycle in the extratropics,

where the strength of the carbon sink is strongest during the warm period (Figure A18). Other statistical metrics suggest a

considerable mismatch between model and reference NEE ($\varepsilon_{rmse} > 1$, $\theta > 1$ and ,$R < 0.2$). The corresponding overall scores

are 0.47 for FLUXNET and 0.53 for CT2019 (Figure 9).

### 3.11   Area burnt and resulting emissions

Wild fires are estimated to burn an area of about $4.6 \times 10^6$ km$^2$ yr$^{-1}$ (ESACCI from 2001-2017 and GFED4S from 2001-

2015), causing emissions of about 2.9 Pg C yr$^{-1}$ (CT2019, 2000-2017) (Table 3, where BURNT refers to area burnt and FIRE

refers to the associated emissions). CLASSIC reproduces the spatial extent of wild fires well considering the uncertainty of the

meteorological forcing, with biases ranging from -28% when forcing CLASSIC with CRUJRAv2, to +14% for GSWP3W5E5.

The larger fractional area burnt values in GSWP3W5E5 are not related to precipitation, as GSWP3W5E5 is the wettest of the

three forcing data sets across all latitudes and seasons. Instead, GSWP3W5E5 has larger values in near-surface windspeeds

compared to the other forcing data sets, which increase the probability of wild fires.

Model values for emissions from wild fires tend to be lower than observed, with biases ranging from -37% (CRUJRAv2)

to -7% (GSWP3W5E5). CLASSIC reproduces the zonal mean pattern of both variables well, with values that are largest at

10°S and 10°N (Figure 12). The sign of the zonal mean bias is consistent between both variables. Biases tend to be negative

in the humid parts of the tropics and positive in the drier parts of the tropics (Figure 12 and A19). CLASSIC reproduces





the seasonality of both variables well, with lower values when the solar zenith angle in the tropics is small (Figure A20 and A21), which is indicative for the position of the intertropical convergence zone. Nevertheless, other statistical metrics suggest that considerable differences between model and reference values remain ($\varepsilon_{rmse} = 1.0$, $\theta > 2$ months, $\varepsilon_{iav} \approx 1$, and $R \leq 0.51$) (Figure 5). The resulting overall scores for fractional area burnt and emissions from fires are 0.51 and 0.47, respectively (Figure 9). These score values are not very sensitive to the choice of forcing ($\Delta S_{overal} \leq 0.01$).

### 3.12 Vegetation biomass

Reference values from GEOCARBON estimate globally accumulated above-ground living biomass (AGB) with 214.5 Pg C. The corresponding values from CLASSIC range from 364.9 Pg C (+70%) to 442.3 Pg C (+106%) (Table 3). The bias is largest between 20°N and 40°N, in particular in the Eastern US and in East Asia (Figure 12 and A22a). Comparing CLASSIC against vegetation carbon (i.e. above and below ground living biomass, CVEG) for the tropics provided by Saatchi et al. (2011) suggests smaller biases (+2% for CRUNCEP to +33% for CRUJRAv2) (Figure 5, 12, and A22b). Spatial correlations are weak, with $R \leq 0.62$ for above ground biomass and $R \leq 0.71$ for vegetation carbon (Figure 5). The corresponding overall scores are 0.55 ($\Delta S_{overall} = 0.01$) for above ground biomass and 0.68 ($\Delta S_{overall} = 0.05$) for vegetation carbon in the tropics (Figure 9).

To investigate differences between CLASSIC and GEOCARBON in greater detail we evaluate model and reference above ground biomass values against forest inventory data (FOS). The comparison shows a larger negative bias for GEOCARBON (-50%) and a smaller negative bias for CLASSIC (-3% for CRUJRAv2 to -21% for CRUNCEP) (Figure 5 and 2d). This finding suggests that GEOCARBON may be subject to considerable uncertainties, which may contribute to the large bias when evaluating CLASSIC against GEOCARBON. More forest inventory data are required to reach a conclusive result, as the current spatial coverage of inventory plots provided by FOS is modest. Evaluating CLASSIC against FOS yields an overall score of 0.64 (Figure 9).

### 3.13 Soil carbon

The global stock of soil carbon (CSOIL) is estimated with 1198 Pg C (HWSD). The corresponding values from CLASSIC range from about 956 Pg C (-20%, CRUNCEP) to 1109 Pg C (-8%, CRUJRAv2) (Table 3). CLASSIC reproduces the zonal mean patterns of soil carbon well, with a tendency for negative biases in the Arctic (-10 kgC m$^{-2}$) (Figure 12). Some of this bias is related to a small number of grid cells at high latitudes with particularly large biases (-40 kgC m$^{-2}$) (Figure A23a). The larger values in the reference data may be related to the presence of permafrost and peatland, which are currently not represented in CLASSIC. The spatial variability of soil carbon is well reproduced ($\sigma \approx 0.92$), but the spatial correlation is rather weak ($R \approx 0.5$) (Figure 5). The corresponding overall score is 0.68 ($\Delta S_{overall} = 0.01$) (Figure 9).

### 3.14 Leaf area index

Evaluating CLASSIC with gridded LAI data from AVHRR (1982-2010) and MODIS (2000-2009) suggests large positive global mean biases that range from +31% (+0.42 m$^{-2}$ m$^{-2}$) to +59% (+0.88 m$^{-2}$ m$^{-2}$), depending on the forcing and reference





data (Figure 5). While CLASSIC reproduces the general pattern of zonal mean LAI reasonably well, the model has a positive bias across most latitudes (Figure 12). However, biases have a local minimum at the equator, especially when using MODIS as a reference. All six evaluations tend towards positive biases in forested regions of the NH extratropics and the drier parts of the

tropics (Figure A23b). In the more humid tropics of the Amazon and Congo basin, biases range from negative to positive values, depending on the forcing and reference data. Hovmoeller diagrams show that the positive bias in the NH extratropics occurs mainly during the cold season, suggesting a possible relation with snow, leaf shedding, and/or the larger solar zenith angle (Figure A24). For most of the remaining months and latitudes, LAI biases range from positive to negative biases, suggesting that modeled values are reasonably accurate when considering the uncertainty of the meteorological forcing and reference data.

Other statistical metrics suggest further differences between model and reference LAI. Values of $\varepsilon_{rmse}$ are larger than for most other variables (0.99 to 1.31) and time lags in seasonal peaks range between one and two months (Figure 5). The spatial variability of LAI is considerably larger for model data than gridded reference data ($\sigma$ ranges from 1.27 to 1.66), and the correlation coefficients are reasonable, but not impressive (0.7 to 0.8). The corresponding overall scores are 0.49 ($\Delta S_{overall} = 0.01$) and 0.53 ($\Delta S_{overall} = 0.03$) when evaluating CLASSIC against gridded LAI reference data from MODIS

and AVHRR, respectively (Figure 9).

To investigate the apparent LAI bias further we evaluate model and reference LAI values with in situ measurements of maximum LAI provided by ORNL. The comparisons show larger negative biases for MODIS (-37%) and AVHRR (-33%), and smaller negative biases for CLASSIC (-19% to -24%) (Figure 5 and 2f for CLASSIC only). This suggests that (i) it is unlikely that CLASSIC overestimates maximum LAI, and (ii) the large positive LAI bias discussed above may in part be due to an

underestimation of LAI by the gridded reference data set. Furthermore, evaluating CLASSIC against in situ LAI observations from CEOS shows mean biases that range from +1% to +22% (Figure 5 and 2e). The corresponding values for MODIS and AVHRR are -6% and -4%, respectively (not shown). Given these findings, we conclude that the large positive LAI found for CLASSIC when evaluated against AVHRR and MODIS must be interpreted with caution. One possible reason for lower LAI in satellite-based reference data at high-latitudes is the problems associated with LAI retrievals during winter months when the

sun is low and the surface is snow covered.

### 3.15  Benchmark scores derived from reference data only

Benchmark scores are produced by comparing two independent reference data sets (Figure 3). This exercise fulfills three purposes. First, benchmark scores quantify observational uncertainty, where lower values imply larger uncertainties. Second, we can evaluate the quality of gridded products using in situ measurements. Third, comparing model scores against benchmark

scores shows how well CLASSIC performs relative to the performance of a reference data set. The comparison of two reference data sets is limited to the time period for which both data sets overlap.

Our results demonstrate that most benchmarks are considerably low, suggesting that observational uncertainties are large. For most variables, model skill scores are very similar to benchmark skill scores, implying that CLASSIC performs similarly to a reference data set. In some cases, CLASSIC even outperforms gridded reference data sets when evaluating gridded reference

data and model output against in situ measurements. The remaining section discusses these findings in more detail.





Contrary to our expectations, most benchmark scores are far below unity (Figure 13). Only two variables show $S_{overall}$ values equal or larger than 0.8 (net surface radiation and net SW radiation). The largest uncertainties are found for NEE, with $S_{overall} = 0.49$ when comparing CT2019 against FLUXNET. Even lower score values are found when evaluating FluxCom NEE against FLUXNET due to the known limitations of the FluxCom data set discussed in section 2.4.2.

Comparing gridded reference data against in situ measurements shows subtle differences in how well gridded products perform for the same variable. For net surface radiation, $S_{overall}$ values range from 0.79 for CLASSr to 0.83 for CERES when evaluated against FLUXNET. For latent heat flux, CLASSr ($S_{overall} = 0.71$) performs slightly better than FluxCom ($S_{overall} = 0.68$). For sensible heat flux, CLASSr and FluxCom perform equally well ($S_{overall} = 0.69$). For snow water equivalent, the blended multi-reference mean data (ECCC; $S_{overall} = 0.66$) performs very similarly to the individual reference data ($S_{overall}$: 

0.64 to 0.66) when evaluated against in situ measurements provided by Mortimer et al. (2020). For GPP, the $S_{overall}$ values are largest for GOSIF (0.66), followed by MODIS (0.65) and FluxCom (0.63) when evaluated against FLUXNET. In the case of NEE, $S_{overall}$ values are larger for FluxCom (0.56) than for CT2019 (0.49). However, it must be noted that the majority of FLUXNET sites are located in the NH extratropics, implying that corresponding scores reflect the performance in primarily non-tropical regions. For LAI, the $S_{overall}$ values are 0.61 for MODIS and 0.60 for AVHRR when evaluated against in situ

measurements from ORNL and 0.74 (MODIS) and 0.77 (AVHRR) when compared against data from CEOS.

In most cases, model skill score values are very similar to the corresponding benchmark score values. For instance, evaluating CERES net surface radiation against FLUXNET yields $S_{overall} = 0.83$. The corresponding mean score value from all three CLASSIC ensemble members is 0.80. For a number of cases, the CLASSIC ensemble performs just as good as independent reference data sets. For instance, $S_{overall}$ values for LW radiation are both 0.8 when evaluating either GEWEXSRB against

CERES or the CLASSIC ensemble against CERES. In three cases (fractional area burnt, LAI, and snow water equivalent) model scores and benchmark scores differ by more than 0.1, which is discussed next.

The $S_{overall}$ benchmark score for fractional area burnt is 0.62 (GFED4S versus ESACCI) and the corresponding model skill score is 0.51 (CLASSIC ensemble versus ESACCI). The score difference is driven by deficiencies in reproducing spatial patterns ($\Delta S_{dist} = -0.27$). For LAI, the $S_{overall}$ benchmark score is 0.75 (AVHRR versus MODIS) and the $S_{overall}$ model

score is 0.49 (CLASSIC ensemble versus MODIS). This difference is primarily driven by lower model skill scores in $S_{bias}$ and $S_{rmse}$. However, in section 3.14 we conclude that LAI biases derived from gridded reference data must be interpreted with caution. Comparing LAI benchmark scores derived from in situ measurements (AVHRR versus ORNL, $S_{overall} = 0.60$) shows much greater similarity with the corresponding model skill scores (CLASSIC ensemble versus ORNL, $S_{overall} = 0.63$). Similar results apply when conducting the same analysis with LAI from MODIS rather than AVHRR.

For snow water equivalent, the $S_{overall}$ benchmark skill score exceeds the corresponding model skill score by 0.11 when evaluating Crocus and the CLASSIC ensemble against GS3. However, using in situ measurements yields benchmark and model skill scores that are more similar. For instance, evaluating the snow water equivalent reference ensemble ECCC and the CLASSIC ensemble against data from Mortimer et al. (2020) yield identical $S_{overall}$ values (0.66).





For some variables, CLASSIC even outperforms gridded reference data when evaluated against in situ observations. This
applies to net surface radiation from CLASSr, latent heat flux from FluxCom, snow water equivalent from Crocus, GS3, and
MERRA2, GPP from FluxCom and MODIS, and LAI from MODIS and AVHRR.

To rank model performance by variable, we compute the average model-benchmark score difference across all evaluations
for each variable. Following this approach, model performance increases as follows, where the average model-benchmark score
differences are provided in parenthesis: fractional area burned (-0.12), runoff (-0.1), soil heat flux (-0.09), LAI (-0.07), net SW
radiation (-0.06), NEE (-0.05), above-ground biomass (-0.03), GPP (-0.03), surface albedo (-0.02), snow water equivalent
(-0.01), net surface radiation (-0.01), sensible heat flux (-0.01), net LW radiation (0.01), latent heat flux (0.03), ecosystem
respiration (0.03). This ranking excludes the FluxCom NEE evaluation against CT2019 due to the unrealistic FluxCom NEE
values in the tropics. To conclude, comparing model skill scores against benchmark scores derived from reference data suggests
that model performance is reasonably good when considering the uncertainties associated with the reference data.

## 4    Discussion

We begin our discussion by comparing our findings with the skill scores presented for CLASS-CTEM by Friedlingstein et al.
(2019). While our results are largely consistent, we can now provide more insights that facilitate the interpretation of those
scores. Our findings show that the poor performance of LAI is mainly due to a large positive bias that occurs in the NH
extratropics during the cold season. An evaluation against in-situ LAI measurements show that biases can be positive, negative,
or close-to-zero, depending on the respective forcing and in situ reference data set. Furthermore, CLASSIC performs very
similar to MODIS and AVHRR when comparing all three data sets against in situ LAI measurements. Also, evaluating MODIS
against AVHRR yields an overall score of 0.75, which implies a considerable mismatch between both reference data sets.
Given the uncertainties in remotely-sensed LAI retrievals discussed in section 2.4.2, it is very well possible that the mismatch
between modeled and gridded reference LAI is, at least to some extent, due to deficiencies in remotely-sensed LAI.

The NEE scores presented in Friedlingstein et al. (2019) are based on in-situ observations and empirically up-scaled eddy
covariance measurements by Jung et al. (2009). The latter presents an earlier version of the FluxCom data set employed in this
study. Either data set has a much stronger carbon sink in the tropics compared to results from inversion models (Jung et al.,
2019b). Potential reasons for this mismatch include ($i$) a sampling bias towards ecosystems with a large carbon sink, ($ii$) miss-
ing predictor variables related to disturbance and site-history, and ($iii$) biases of eddy covariance NEE measurements, possibly
due to night-time advection of carbon dioxide under tall tropical forest canopies (Jung et al., 2019b). Furthermore, the inter-
annual variability of FluxCom data is considerably lower compared to results from inversion methods. Evaluating FluxCom
and CLASSIC against the inversion model results from CT2019 shows that the model outperforms FluxCom. Furthermore,
the NEE skill scores of CT2019 and CLASSIC are similar when assessing both data sets against FLUXNET (0.49 and 0.47,
respectively). We conclude that the low NEE score presented in Friedlingstein et al. (2019) is mainly a sign of observational
uncertainties rather than model deficiencies.





Our benchmarking approach is subject to limitations, which are discussed next. Evaluating CLASSIC against eddy covariance measurements has numerous short-comings. First, eddy covariance measurements from FLUXNET lack energy balance closure, where sensible and latent heat fluxes are about 20% lower compared to the available energy (Wilson et al., 2002). Second, FLUXNET data are subject to gaps, which often occur when conditions are unsuitable for making measurements.

Third, eddy covariance measurements have a much smaller foot print size (Schmid, 1994) compared to our grid cell size of $2.8125° \times 2.8125°$. The spatial mismatch implies that the measurements are not necessarily representative for the area covered by the grid cell. Considerable differences between measured and modeled values are, therefore, expected. However, CLASSIC can also be run at the site level using local conditions (Melton et al., 2020). The resulting values for $S_{overall}$ are similar to the ones presented here, which suggests that a comparison at the grid cell level is still meaningful. Finally, the periods of

evaluation differ among reference data sets. For instance, LAI data from MODIS and AVHRR cover the periods 2000-2017 and 1982-2010, respectively.

Three variables assessed in this study were evaluated against one reference data set only, namely soil moisture, $CO_2$ emissions from fires, and soil carbon. It remains, therefore, unclear to what extent model performance for those three variables is affected by observational uncertainty. Future evaluations will, therefore, consider additional reference data sets, including soil

carbon from the World Soil Information Service (Batjes et al., 2017), as well as additional in situ measurements for surface radiation provided by the Baseline Surface Radiation Network (Driemel et al., 2018). Future developments of AMBER could introduce a new score that quantifies how well a model reproduces the sensitivity of ecosystems to climate extremes, which are shown to affect terrestrial ecosystems in profound ways (Reichstein et al., 2013).

## 5   Conclusions

This study assesses the energy, water, and carbon cycle simulated by CLASSIC, using a wide range of in situ measurements and globally gridded reference data. We account for uncertainties in the meteorological forcing by evaluating an ensemble of three model runs that are produced with different forcing data sets. Uncertainties in the reference data are accounted for by comparing model scores against benchmark scores, which are based on reference data only. The method has been implemented in a new R package referred to as AMBER, which is publicly available at the Comprehensive R Archive Network (https:

//CRAN.R-project.org/package=amber).

The complexity of LSMs and increasing availability of global earth observations demand for comprehensive methods of model evaluation. Such methods must account for uncertainties associated with model inputs and observation-based reference data to yield robust results. A centralized and updated reference database that includes spatially explicit information of uncertainties would be highly valuable.

Our results show that model performance is very sensitive to the choice of meteorological forcing and reference data. For instance, for about half of all variables assessed in this study, the sign of the bias changes depending on what forcing and reference data are used. Comparing model and benchmark scores confirms that CLASSIC performance is strongly affected by





observational uncertainty. Taking this uncertainty into account shows that CLASSIC performs very well across a wide range of variables. Our results will serve as a baseline for guiding and monitoring future CLASSIC development.

*Code and data availability.*  The data, scripts, computational environment, and instructions required for reproducing the results presented in this paper can be downloaded from https://doi.org/10.5281/zenodo.4010681. AMBER (version 1.0.3) and its documentation are also available through the Comprehensive R Archive Network (CRAN; https://CRAN.R-project.org/package=amber). The full set of Figures produced by AMBER for this study can be accessed at https://cseiler.shinyapps.io/ShinyAmber. The CLASSIC code and its software container are available at https://doi.org/10.5281/zenodo.3522407.

**Appendix A**

The computation of GPP is based on the parameterizations by Farquhar et al. (1980), Collatz et al. (1991) and Collatz et al. (1992), with some minor adjustments documented in Arora and Boer (2003). The gross leaf photosynthesis rate ($G_o$) is limited by the availability of photosynthetically active radiation (PAR) ($J_e$), Rubisco ($J_c$), and by the transport capacity ($J_s$) (Melton and Arora, 2016). To compute GPP, $G_o$ is first downregulated in response to nutrient availability and then multiplied with the

fraction of photosynthetically active radiation $f_{PAR}$:

$$f_{PAR} = \frac{1}{k_n} \left(1 - \exp(-k_n \Lambda)\right), \tag{A1}$$

where $k_n$ is the extinction coefficient that describes the nitrogen and time-mean PAR profile along the depth of the canopy and $\Lambda$ is the leaf area index (Arora et al., 2009). $\Lambda$ is computed as the product of specific leaf area (SLA) and leaf biomass ($C_L$), where

$$SLA = 25.0 \Upsilon_L^{-0.5} \tag{A2}$$

and $\Upsilon_L$ being a PFT-specific leaf life span measured in years.

Ecosystem respiration is the sum of autotrophic ($R_a$) and heterotrophic respiration ($R_h$). Autotrophic respiration is the sum of maintenance ($R_m$) and growth respiration ($R_g$). Maintenance respiration is divided into respiration associated with the maintenance of leaves ($R_{mL}$), stems ($R_{mS}$), and roots ($R_{mR}$). Leaf maintenance respiration is scaled from the leaf to the

canopy level using $f_{PAR}$, and growth respiration ($R_g$) is estimated as a fraction of GPP. Heterotrophic respiration is the sum of respiration from litter and soil carbon, and is affected by soil temperature and soil moisture.

The reflectivity of plant surfaces varies considerably within the shortwave spectrum, with higher absorptivity in the visible range due to photosynthesis and lower absorptivity in the near-infrared (IR) range. CLASSIC, therefore, computes surface albedo for the visible and near-IR spectrum separately. Both approaches are identical, except for the use of different constants

that affect the background albedos for soil, vegetation, and snow, as well as canopy transmissivities. The surface albedo for the visible spectrum is the sum of the visible albedos of ($i$) bare ground ($\alpha_{vs,g}$), ($ii$) open snow ($\alpha_{vs,sn}$), ($iii$) vegetation over





bare ground ($\alpha_{vs,cn}$), and ($iv$) vegetation over snow ($\alpha_{vs,cs}$), multiplied with the corresponding subarea fractional coverage ($f_g$, $f_{gs}$, $f_c$, and $f_{cs}$, respectively; Figure A25):

$$\alpha_{vs} = f_c\,\alpha_{vs,cn} + f_g\,\alpha_{vs,g} + f_{cs}\,\alpha_{vs,cs} + f_{gs}\,\alpha_{vs,sn}. \tag{A3}$$

The visible albedo of bare ground ($\alpha_{vs,g}$) ranges from the visible albedo of wet soil ($\alpha_{g,w,v}$) to the visible albedo of dry soil ($\alpha_{g,d,v}$), depending on the soil moisture content in the first soil layer ($\theta_{l,1}$). The visible albedo of open snow ($\alpha_{vs,sn}$) depends on the all-wave albedo of the snow pack ($\alpha_{sno}$). The visible albedo of vegetation over bare ground ($\alpha_{vs,cn}$) is the product of the total vegetation albedo of a PFT over bare soil ($\alpha_{sn,n}$) and its corresponding fractional coverage ($f_{can}$). The former depends on the canopy albedo ($\alpha_{vs,cx}$), the visible albedo of ground under vegetation canopy ($\alpha_{vs,gc}$), the sky view factor ($\chi$), and the
visible transmissivity of the vegetation ($\tau_{vs}$):

$$\alpha_{vs,n} = (1 - \chi)\alpha_{vs,cx} + \chi\tau_{vs}\alpha_{vs,gc}. \tag{A4}$$

The visible albedo of ground under vegetation canopy ($\alpha_{vs,gc}$) is set equal to the visible albedo of bare ground ($\alpha_{vs,g}$) computed above. The canopy albedo $\alpha_{vs,cx}$ is a function of the fractional coverage of canopy by frozen water ($f_{snowc}$) and the average background visible albedo of snow covered vegetation ($\alpha_{vs,wc}$):

$$\alpha_{vs,cx} = f_{snow,c}\,\alpha_{vs,wc} + (1.0 - f_{snow,c})\,\alpha_{vs,c}. \tag{A5}$$

The visible transmissivity of the vegetation ($\tau_{vs}$) follows Beer's law of radiation transfer:

$$\tau_{vs} = \exp(-\kappa\,\Lambda_{pai}), \tag{A6}$$

where $\kappa$ is the canopy extinction coefficient, which depends on the cloud fraction, the solar zenith angle, and PFT-specific constants. The sky view factor ($\chi$) expresses the degree of closure of a canopy, and is a function of a PFT-specific constant ($c$)
and the plant area index ($\Lambda_{pai}$), which is the sum of $\Lambda$ and stem area index:

$$\chi = \exp(c\,\Lambda_{pai}). \tag{A7}$$

The same approach described for computing $\alpha_{vs,cn}$ applies to the visible albedo of vegetation over snow ($\alpha_{vs,cs}$), except that the terms $f_{can}\alpha_{vs,n}$ and $\alpha_{vs,gc}$ are replaced with the corresponding terms for canopy over snow, i.e. $f_{cans}\alpha_{vs,s}$ and $\alpha_{vs,sc}$.

The visible and near-IR albedos and canopy transmissivities obtained above are now used to compute the net SW radiation for
the canopy ($K_{*,c}$) and underlying ground (or snow) ($K_{*,g}$):

$$K_{*,g} = \hat{\tau}_{vs}K^{\downarrow}_{vs}(1 - \alpha_{vs,g}) + \hat{\tau}_{nir}K^{\downarrow}_{nir}(1 - \alpha_{nir,g}) \text{ and } K_{*,c} = K^{\downarrow}_{vs}(1 - \alpha_{vs,c}) + K^{\downarrow}_{nir}(1 - \alpha_{nir,c}) - K_{*,g}, \tag{A8}$$

where $K^{\downarrow}$ is the downwelling SW radiation, carets denote values that are averaged over multiple PFTs, and the suffix $nir$ indicates that the respective variable applies to the near-IR spectrum (Figure A26). The net SW radiation is then the sum of $K_{*g}$, $K_{*c}$, and the SW radiation transmitted into snow, if present ($Q_{trans}$).
The net LW radiation for the canopy ($L_{*,g}$) and underlying ground (or snow) ($L_{*,c}$) follow the Stefan-Boltzmann law:

$$L_{*,g} = (1 - \hat{\chi})\sigma\bar{T}_c^4 + \hat{\chi}L^{\downarrow} - \sigma T(0)^4 \text{ and } L_{*,c} = (1 - \hat{\chi})\left(L^{\downarrow} + \sigma T(0)^4 - 2\sigma\bar{T}_c^4\right), \tag{A9}$$





where $L^{\downarrow}$ is the downwelling LW radiation, $\sigma$ is the Stefan-Boltzmann constant, $\bar{T}_c$ is the effective canopy temperature, and $T(0)$ is the temperature of the underlying ground (Figure A26).

The sensible heat flux from the ground ($Q_{H,g}$) and canopy $Q_{H,c}$ are given as:

$$Q_{H,g} = \rho_a c_p (T(0)_{pot} - T_{a,c})/r_{a,g} \text{ and } Q_{H,c} = \rho_a c_p (\bar{T}_c - T_{a,c})/r_b, \tag{A10}$$

where $\rho_a$ is the air density, $c_p$ is the specific heat capacity, $T(0)_{pot}$ is the potential air temperature at the ground, $T_{a,c}$ is the canopy air temperature, $r_{a,g}$ is the surface resistance, and $r_b$ is the leaf boundary resistance. Air density is a function of air temperature and pressure as defined in the ideal gas law. The resistances $r_{a,g}$ and $r_b$ are defined as:

$$\frac{1}{r_{a,g}} = 1.9 \times 10^{-3} (T(0)_v - T_{ac,v})^{1/3} \text{ and } \frac{1}{r_b} = v_{ac}^{0.5} \sum f_i \gamma_i \Lambda_i^{0.5}/0.75[1 - \exp(-0.75\Lambda_i^{0.5})], \tag{A11}$$

where $T(0)_v$ is the virtual potential temperature of the surface, $T_{ac,v}$ is the virtual temperature of the air in the canopy space, $v_{ac}$ is the wind speed in the canopy air space, $f_i$ is the fractional coverage of each vegetation type $i$ over the subarea in question, and $\gamma_i$ is a vegetation-dependent parameter that incorporates the effects of leaf dimension and sheltering.

The latent heat flux from the ground ($Q_{E,g}$) and canopy ($Q_{E,c}$) are:

$$Q_{E,g} = L_v \rho_a (q(0) - q_{a,c})/r_{a,g} \text{ and } Q_{E,c} = L_v \rho_a (q_c - q_{a,c})/(r_b + r_c), \tag{A12}$$

where $L_v$ is the latent heat of vaporization, $q(0)$ is the specific humidity at surface, $q_{a,c}$ is the specific humidity of air within vegetation canopy space, $q_c$ is the saturated specific humidity at canopy temperature, and $r_c$ is the stomatal resistance to transpiration. In the absence of water stress, the unstressed $r_c$ is a function of $K^{\downarrow}$, $\Lambda$, and $\kappa$. If soil moisture is limiting transpiration, $r_c$ depends on air temperature, vapor pressure deficit, and a soil moisture suction coefficient.

The soil heat flux ($G(0)$) is estimated from the vertical soil temperature gradient and the thermal conductivity of the first three
soil layers:

$$G(0) = -\lambda_t \frac{\partial T}{\partial z}. \tag{A13}$$

The same approach is applied in the presence of snow, where $\lambda_t$ is replaced with the thermal conductivity of snow ($\lambda_s$) and $T$ becomes the snow temperature ($T_s$). The latter depends on the snow heat capacity, which is a function of snow water content. The surface and canopy energy balance are then expressed as:

$$K_{*,g} + L_{*,g} - Q_{H,g} - Q_{E,g} - G(0) = 0 \text{ and } K_{*,c} + L_{*,c} - Q_{H,c} + Q_{H,g} - Q_{E,c} - \Delta Q_{S,c}) = 0, \tag{A14}$$

where $\Delta Q_{S,c}$ represents the change of energy storage in the canopy. In practice, the terms on the left-hand side don't necessarily add up to zero. CLASSIC attempts to approximate a zero net balance by slightly adjusting $T(0)$ until the absolute residual of the surface energy balance is less than 5.0 W m$^{-2}$ , or until the absolute value of the iteration step most recently used is less than 0.01 K.




*Author contributions.* C.S. developed AMBER, conducted the analysis, and wrote the first draft of the manuscript. J.M. conducted the CLASSIC experiments assessed in this study and contributed to the methods that address uncertainties in the meteorological forcing and reference data. V.A. created CTEM. L.W. processed albedo and leaf area index MODIS data. All authors contributed to the final version of the manuscript.

*Competing interests.* The authors declare no competing interests.

*Acknowledgements.* The authors wish to thank all groups that provided public access to the reference data listed in Table 1. The eddy covariance data that are shared by the FLUXNET community include the networks AmeriFlux, AfriFlux, AsiaFlux, CarboAfrica, CarboEuropeIP, CarboItaly, CarboMont, ChinaFlux, Fluxnet-Canada, GreenGrass, ICOS, KoFlux, LBA, NECC, OzFlux-TERN, TCOS-Siberia, and USCCC. The FLUXNET eddy covariance data processing and harmonization was carried out by the European Fluxes Database Cluster, AmeriFlux Management Project, and Fluxdata project of FLUXNET, with the support of CDIAC and ICOS Ecosystem Thematic Center, and the OzFlux, ChinaFlux and AsiaFlux offices. Dr. Colleen Mortimer compiled in situ measurements of snow water equivalent.



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





**Table 1.** Observation-based reference data used for model evaluation. Meanings of acronyms are provided in the Methods section.

| Source | Variables | Approach ($n$ sites) | Period | Reference |
|---|---|---|---|---|
| **In situ measurements** | | | | |
| FLUXNET2015 | RNS, HFLS, HFSS, HFG | eddy covariance (204) | 1997-2014 | Pastorello et al. (2017) |
| | GPP, RECO, NEE | | | |
| GRDC | MRRO | gauge records (50) | 1980-2010 | Dai and Trenberth (2002) |
| Mortimer | SNW | gravimetry (3271) | 1970-2017 | Mortimer et al. (2020) |
| CEOS | LAI | transfer function (141) | 1999-2017 | Garrigues et al. (2008) |
| ORNL DAAC | LAI | maximum LAI (2653) | 1932-2011 | Iio and Ito (2014) |
| FOS | AGB | allometry (274) | 1999-2018 | Schepaschenko et al. (2019) |
| **Globally gridded data sets** | | | | |
| MODIS | ALBS | BRDF | 2000-2014 | Strahler et al. (1999) |
| | GPP | light use efficiency model | 2000-2016 | Zhang et al. (2017) |
| | LAI | radiative transfer model | 2000-2017 | Myneni et al. (2002) |
| AVHRR | LAI | artificial neural network | 1982-2010 | Claverie et al. (2016) |
| CERES | ALBS, RSS, RLS, RNS | radiative transfer model | 2000-2012 | Kato et al. (2013) |
| GEWEXSRB | ALBS, RSS, RLS, RNS | radiative transfer model | 1984-2007 | Stackhouse et al. (2011) |
| GOSIF | GPP | statistical model | 2000-2017 | Li and Xiao (2019) |
| FluxCom | RNS, HFLS, HFSS | machine learning | 2001-2013 | Jung et al. (2019a) |
| FluxCom | GPP, RECO, NEE | machine learning | 1980-2013 | Jung et al. (2019b) |
| CLASSr | RNS, HFLS, HFSS, HFG, MRRO | blended product | 2003-2009 | Hobeichi et al. (2019) |
| ESA | MRSLL | land surface model | 1979-2017 | Liu et al. (2011) |
| ECCC | SNW | blended product | 1981-2017 | Mudryk (2020) |
| Brown | SNW | temperature index model | 1981-2017 | Brown et al. (2003) |
| Crocus | SNW | snow pack model | 1981-2017 | Brun et al. (2013) |
| GS3 | SNW | passive microwave | 1981-2017 | Takala et al. (2011) |
| MERRA2 | SNW | reanalysis | 1981-2017 | Gelaro et al. (2017) |
| GFED4S | BURNT | burned-area mapping | 2001-2015 | Giglio et al. (2010) |
| ESACCI | BURNT | burned-area mapping | 2001-2017 | Chuvieco et al. (2018) |
| CT2019 | NEE, FIRE | inversion model | 2000-2017 | Jacobson et al. (2020) |
| GEOCARBON | AGB | machine learning | NA | Avitabile et al. (2016), |
| | | | | Santoro et al. (2015) |
| Saatchi | CVEG | machine learning | 2000 | Saatchi et al. (2011) |
| HWSD | CSOIL | soil inventory | NA | Wieder (2014) |
| | | | | Todd-Brown et al. (2013) |





**Table 2.** Qualitative assessment of global mean biases. Variables for which positive and negative mean biases are found are denoted with $\pm$. Biases that have a consistent sign across all evaluations but are smaller than $\pm 5\%$ are marked with $\leq 5\%$. Larger systematic biases are marked with $+$ or $-$. Numbers provided for each in situ reference data refer to the number of grid cells involved.

| Variable | Acronym | Grid | In situ | Gridded reference data | In situ reference data ($n$) |
|---|---|---|---|---|---|
| Net surface radiation | RNS | $\pm$ | $\pm$ | CERES, CLASSr, FluxCom, GEWEXSRB | FLUXNET(73) |
| Sensible heat flux | HFSS | $\pm$ | $\pm$ | CLASSr, FluxCom | FLUXNET (65) |
| Snow water equivalent | SNW | $\pm$ | $\pm$ | Brown, Crocus, GS3, MERRA2 | Mortimer (363) |
| Runoff | MRRO | $\leq 5\%$ | $\pm$ | CLASSr | GRDC (50 basins) |
| Net LW radiation | RLS | $\pm$ | | CERES, GEWEXSRB | |
| Soil moisture | MRSLL | $\pm$ | | ESA | |
| Fractional area burnt | BURNT | $\pm$ | | ESACCI, GFED4S | |
| Net SW radiation | RSS | $\leq 5\%$ | | CERES, GEWEXSRB | |
| Vegetation carbon in the tropics | CVEG | $\leq 5\%$ | | Saatchi | |
| Latent heat flux | HFLS | $\pm$ | $-$ | CLASSr, FluxCom | FLUXNET (65) |
| Gross primary productivity | GPP | $\pm$ | $-$ | FluxCom, GOSIF, MODIS | FLUXNET (89) |
| Net ecosystem exchange | NEE | $\leq 5\%$ | $+$ | FluxCom, CT2019 | FLUXNET (91) |
| Ecosystem respiration | RECO | $+$ | $\leq 5\%$ | FluxCom | FLUXNET (88) |
| Above ground biomass | AGB | $-$ | $\leq 5\%$ | GEOCARBON | FOS (56) |
| Leaf area index | LAI | $+$ | $\pm$ | AVHRR, MODIS | CEOS (36), ORNL (330) |
| Surface albedo | ALBS | $+$ | | CERES, GEWEXSRB, MODIS | |
| Emissions from wild fires | FIRE | $-$ | | CT2019 | |
| Soil carbon | CSOIL | $-$ | | HWSD | |
| Soil heat flux | HFG | $-$ | $-$ | CLASSr | FLUXNET (60) |





**Table 3.** Globally summed mean values and corresponding biases

| Variable | Ref. ID | Model ID | Ref. | Model | Bias | Bias (%) | Unit | Period |
|----------|---------|----------|------|-------|------|----------|------|--------|
| GPP | FluxCom | CLASSIC.CRUJRAv2 | 115.41 | 131.11 | 15.70 | 13.60 | PgC yr$^{-1}$ | 1980-2013 |
| GPP | FluxCom | CLASSIC.GSWP3W5E5 | 115.31 | 124.23 | 8.92 | 7.74 | PgC yr$^{-1}$ | 1980-2013 |
| GPP | FluxCom | CLASSIC.CRUNCEP | 115.39 | 119.87 | 4.48 | 3.88 | PgC yr$^{-1}$ | 1980-2013 |
| GPP | GOSIF | CLASSIC.CRUJRAv2 | 130.01 | 127.84 | -2.17 | -1.67 | PgC yr$^{-1}$ | 2000-2017 |
| GPP | GOSIF | CLASSIC.GSWP3W5E5 | 129.61 | 120.02 | -9.59 | -7.40 | PgC yr$^{-1}$ | 2000-2016 |
| GPP | GOSIF | CLASSIC.CRUNCEP | 129.73 | 116.60 | -13.13 | -10.12 | PgC yr$^{-1}$ | 2000-2016 |
| GPP | MODIS | CLASSIC.CRUJRAv2 | 121.47 | 133.07 | 11.60 | 9.55 | PgC yr$^{-1}$ | 2000-2016 |
| GPP | MODIS | CLASSIC.GSWP3W5E5 | 121.33 | 125.29 | 3.96 | 3.26 | PgC yr$^{-1}$ | 2000-2016 |
| GPP | MODIS | CLASSIC.CRUNCEP | 121.46 | 121.74 | 0.28 | 0.23 | PgC yr$^{-1}$ | 2000-2016 |
| NEE | CT2019 | CLASSIC.CRUJRAv2 | -4.15 | -4.19 | -0.04 | -0.96 | PgC yr$^{-1}$ | 2000-2017 |
| NEE | CT2019 | CLASSIC.GSWP3W5E5 | -4.18 | -4.36 | -0.18 | -4.31 | PgC yr$^{-1}$ | 2000-2016 |
| NEE | CT2019 | CLASSIC.CRUNCEP | -4.19 | -4.60 | -0.41 | -9.79 | PgC yr$^{-1}$ | 2000-2016 |
| NEE | FluxCom | CLASSIC.CRUJRAv2 | -24.88 | -3.54 | 21.34 | 85.77 | PgC yr$^{-1}$ | 1980-2013 |
| NEE | FluxCom | CLASSIC.GSWP3W5E5 | -24.85 | -3.87 | 20.98 | 84.43 | PgC yr$^{-1}$ | 1980-2013 |
| NEE | FluxCom | CLASSIC.CRUNCEP | -24.88 | -4.13 | 20.75 | 83.40 | PgC yr$^{-1}$ | 1980-2013 |
| FIRE | CT2019 | CLASSIC.CRUJRAv2 | 2.88 | 1.82 | -1.06 | -36.81 | PgC yr$^{-1}$ | 2000-2017 |
| FIRE | CT2019 | CLASSIC.GSWP3W5E5 | 2.90 | 2.70 | -0.20 | -6.90 | PgC yr$^{-1}$ | 2000-2016 |
| FIRE | CT2019 | CLASSIC.CRUNCEP | 2.90 | 2.16 | -0.74 | -25.52 | PgC yr$^{-1}$ | 2000-2016 |
| BURNT | ESACCI | CLASSIC.CRUJRAv2 | 4.61 | 3.31 | -1.30 | -28.20 | 10$^6$ km$^2$ yr$^{-1}$ | 2001-2017 |
| BURNT | ESACCI | CLASSIC.GSWP3W5E5 | 4.62 | 5.27 | 0.65 | 14.07 | 10$^6$ km$^2$ yr$^{-1}$ | 2001-2016 |
| BURNT | ESACCI | CLASSIC.CRUNCEP | 4.62 | 4.05 | -0.57 | -12.34 | 10$^6$ km$^2$ yr$^{-1}$ | 2001-2016 |
| BURNT | GFED4S | CLASSIC.CRUJRAv2 | 4.56 | 3.32 | -1.24 | -27.19 | 10$^6$ km$^2$ yr$^{-1}$ | 2001-2015 |
| BURNT | GFED4S | CLASSIC.GSWP3W5E5 | 4.56 | 5.29 | 0.73 | 16.01 | 10$^6$ km$^2$ yr$^{-1}$ | 2001-2015 |
| BURNT | GFED4S | CLASSIC.CRUNCEP | 4.56 | 4.06 | -0.50 | -10.96 | 10$^6$ km$^2$ yr$^{-1}$ | 2001-2015 |
| AGB | GEOCARBON | CLASSIC.CRUJRAv2 | 214.53 | 442.25 | 227.72 | 106.15 | PgC | 1980-2017 |
| AGB | GEOCARBON | CLASSIC.GSWP3W5E5 | 214.50 | 384.31 | 169.81 | 79.17 | PgC | 1980-2017 |
| AGB | GEOCARBON | CLASSIC.CRUNCEP | 214.53 | 364.94 | 150.41 | 70.11 | PgC | 1980-2017 |
| CSOIL | HWSD | CLASSIC.CRUJRAv2 | 1199.32 | 1108.79 | -90.53 | -7.55 | PgC | 1980-2017 |
| CSOIL | HWSD | CLASSIC.GSWP3W5E5 | 1198.49 | 1023.48 | -175.01 | -14.60 | PgC | 1980-2017 |
| CSOIL | HWSD | CLASSIC.CRUNCEP | 1198.26 | 955.57 | -242.69 | -20.25 | PgC | 1980-2017 |

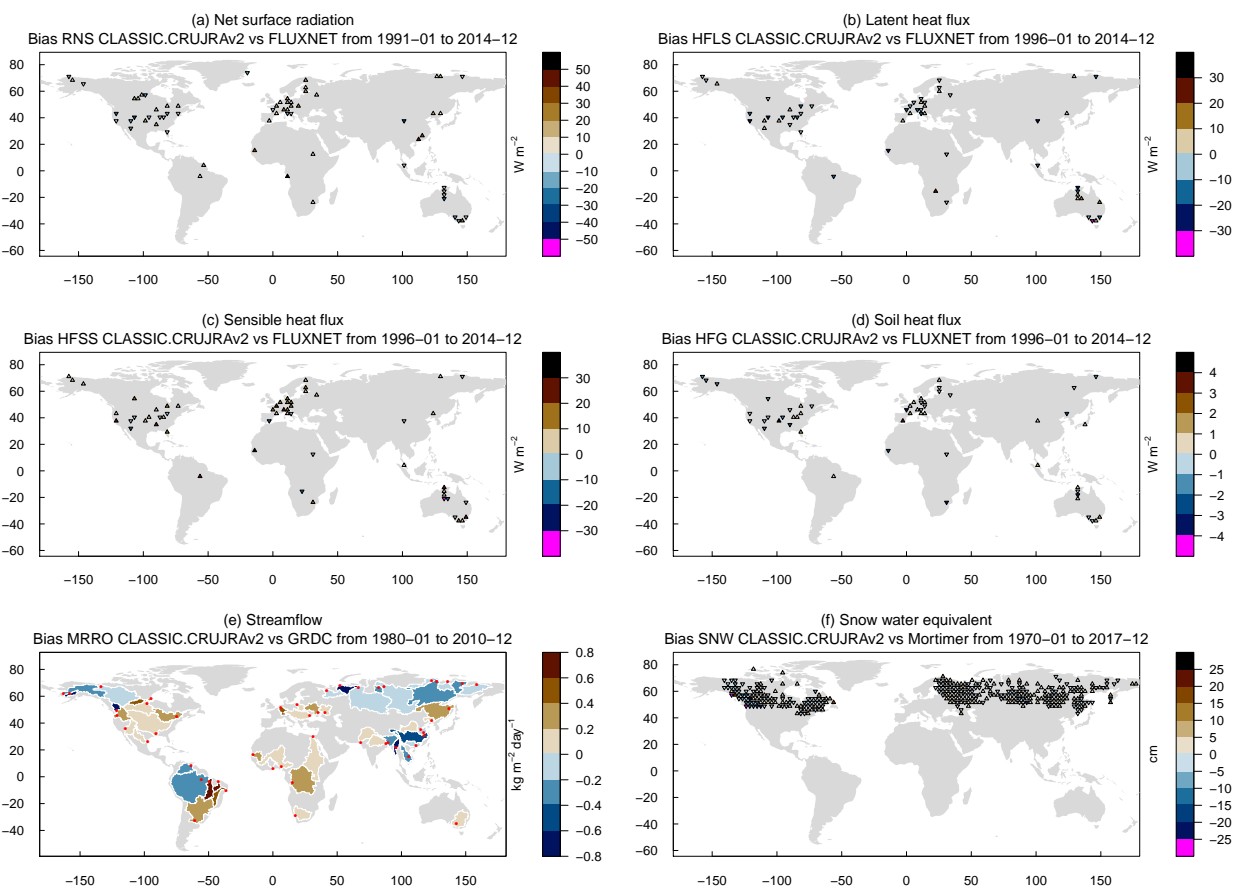

**Figure 1.** Biases when comparing CLASSIC forced with CRUJRAv2 against in situ measurements of (a) net surface radiation, (b) latent heat flux, (c) sensible heat flux, (d) soil heat flux, (e) streamflow, and (f) snow water equivalent. Measurements located in the same model grid cell were merged. Upward and downward pointing triangles indicate positive and negative biases, respectively.

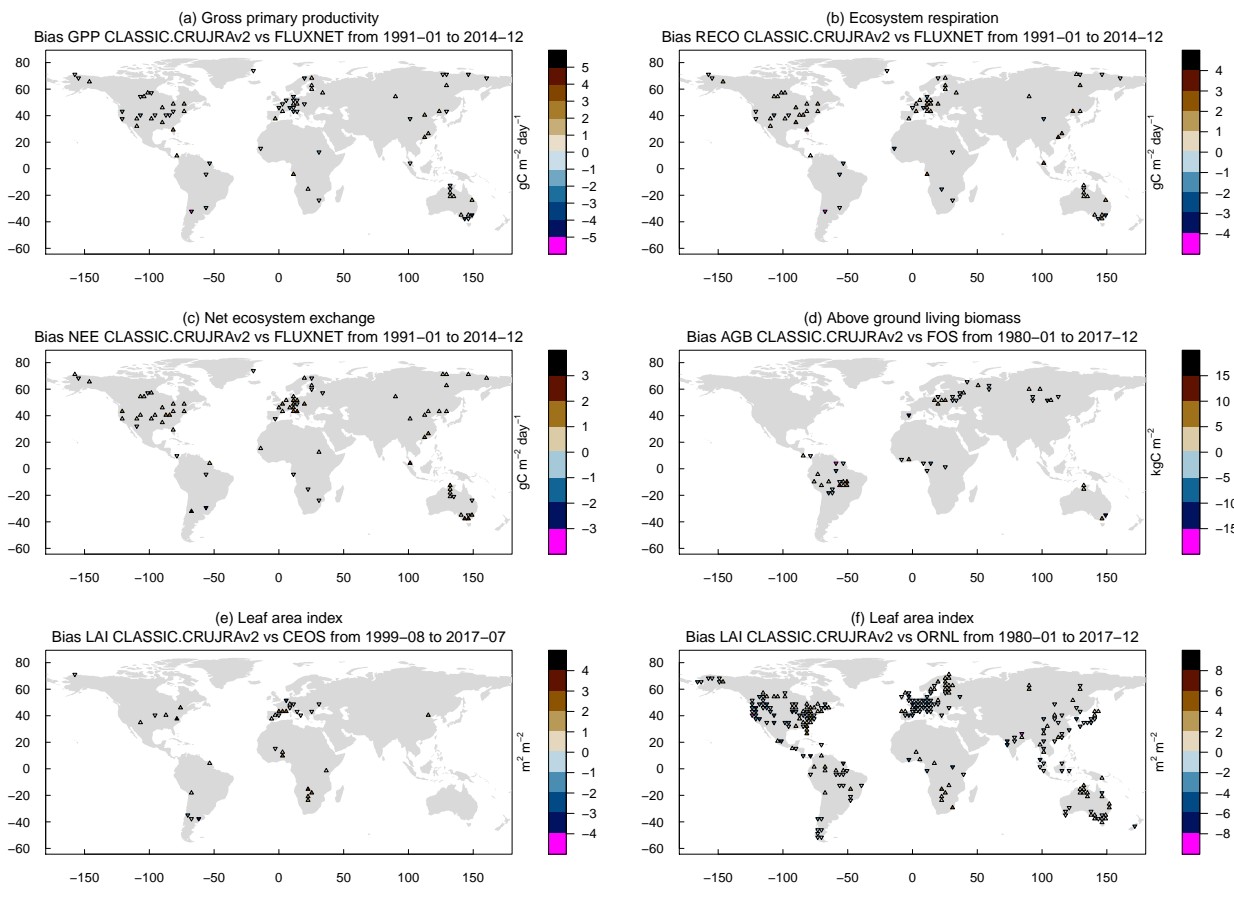

**Figure 2.** Same as Figure 1 but for (a) gross primary productivity, (b) ecosystem respiration, (c) net ecosystem exchange, (d) above-ground biomass, (e) leaf area index, and (f) maximum leaf area index.





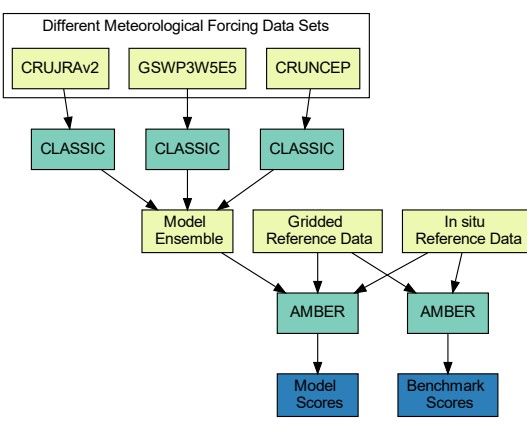

**Figure 3.** Experimental set-up, where yellow boxes refer to data, green boxes to algorithms, and blue boxes to analysis output.





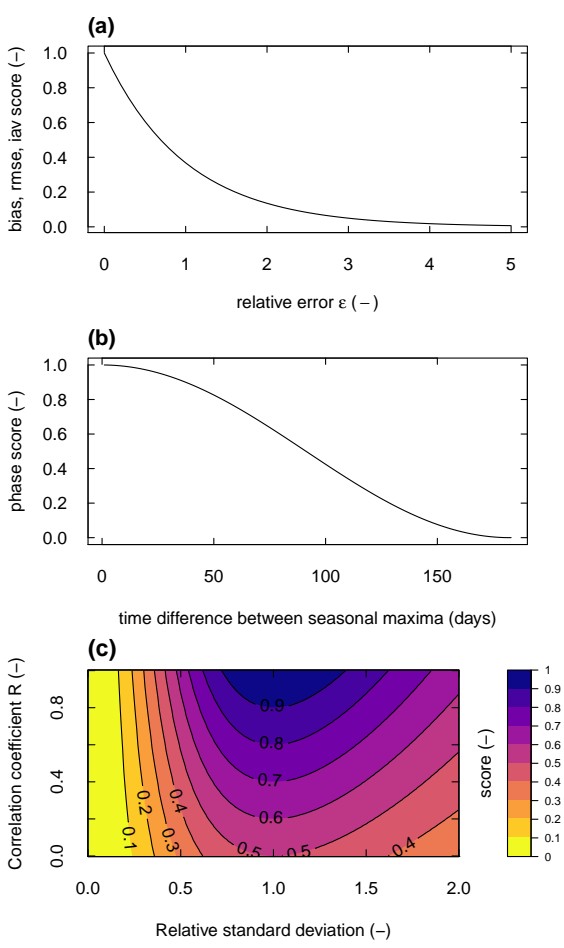

**Figure 4.** Score functions for (a) bias, root-mean-square-error, and inter-annual variability, (b) seasonality, and (c) spatial distribution







**Figure 5.** Global mean values of statistical metrics defined in section 2. The color ranges span the minimum and maximum for each metric. The physical units of the mean and absolute bias are given by the vertical axis on the right-hand-side.



**Figure 6.** (a) Zonal mean values and (b) corresponding biases for variables related to the radiation cycle.



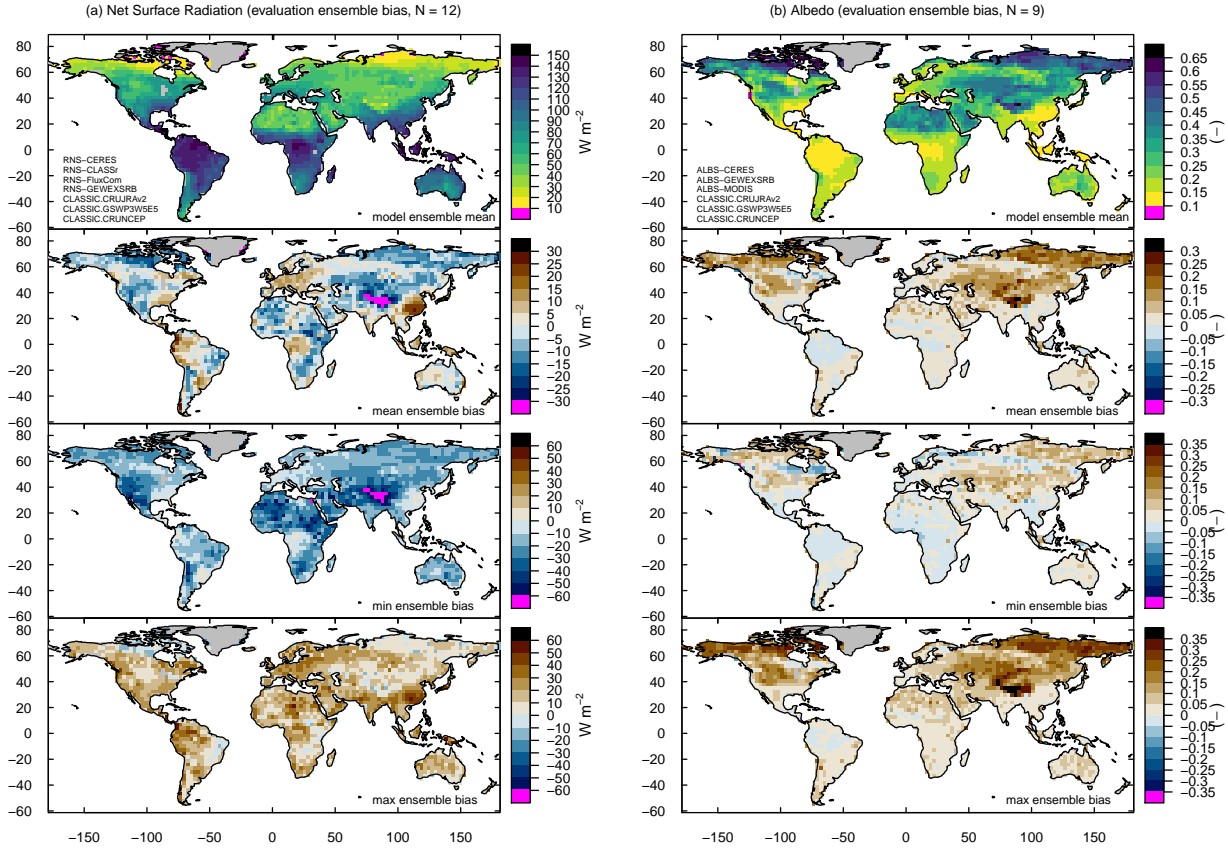

**Figure 7.** Model ensemble mean values and respective biases for (a) net surface radiation and (b) surface albedo. The mean, minimum, and maximum ensemble bias refer to the mean, minimum, and maximum values across all evaluations, respectively. For instance, a minimum bias that is negative implies that at least one evaluation yields a negative bias.



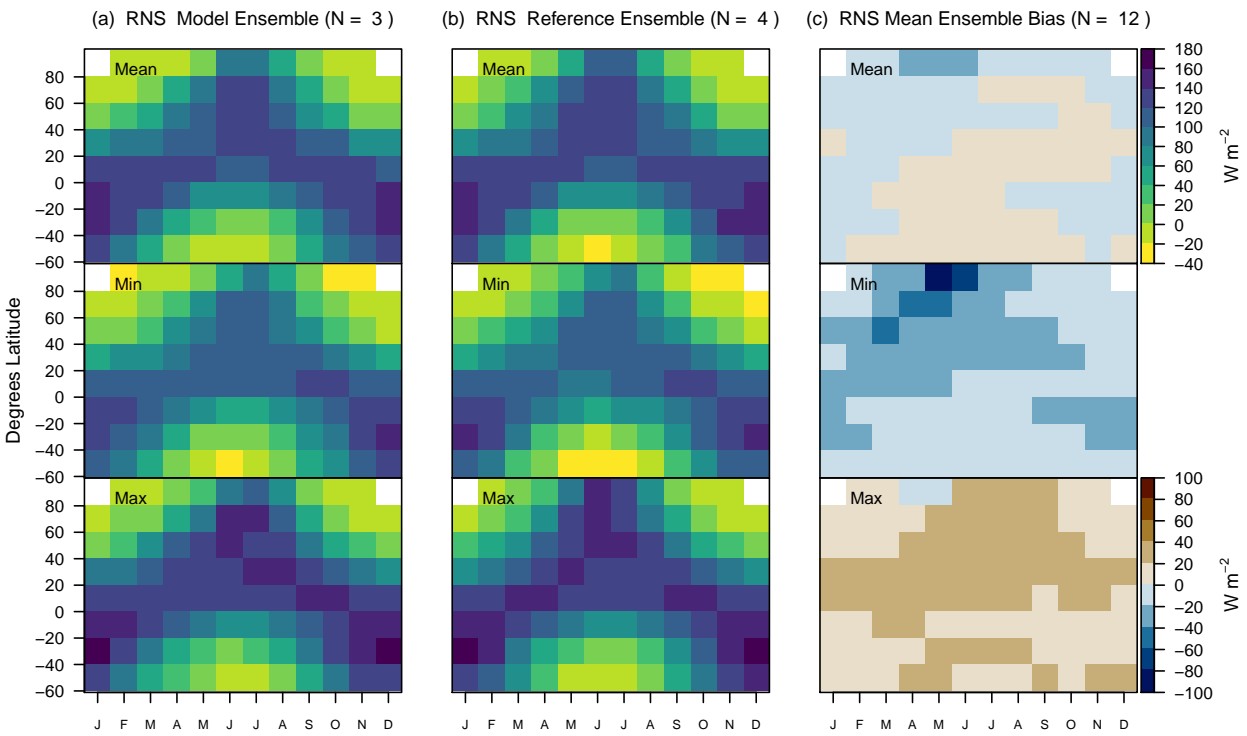

**Figure 8.** Net surface radiation computed from three model runs (CLASSIC driven with CRUJRAv2, GSWP3W5E5, and CRUNCEP) and four reference data sets (CERES, CLASSr, FluxCom, and GEWEXSRB). The top, center, and bottom row provide ensemble mean, minimum, and maximum values for (a) CLASSIC, (b) reference data, and (c) corresponding biases. The mean, minimum, and maximum ensemble bias refer to the mean, minimum, and maximum values across all evaluations, respectively. For instance, a minimum bias that is negative implies that at least one evaluation yields a negative bias.





**Figure 9.** (a) Mean ensemble score, (b) maximum score difference among ensemble members, and ensemble member with the (c) largest and (d) lowest score. Ensemble member IDs refer to model runs conducted with data from (1) CRUJRAv2, (2) GSWP3W5E5, and (3) CRUNCEP.



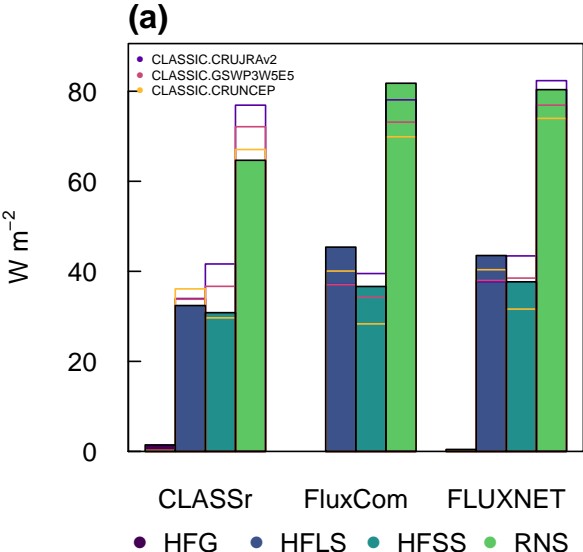

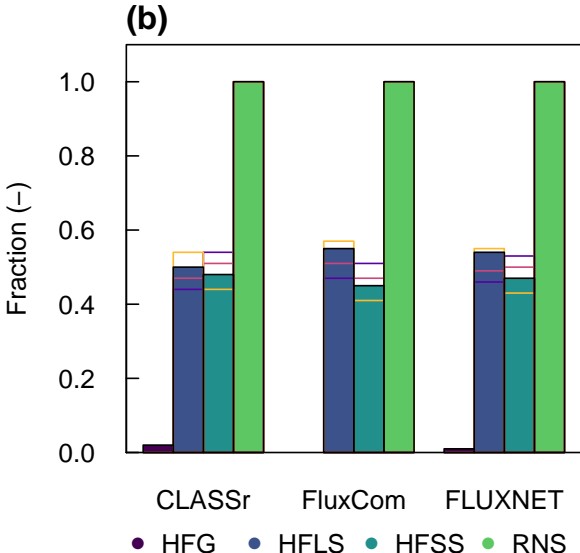

**Figure 10.** (a) Annual mean values of soil heat flux (HFG), latent heat flux (HFLS), sensible heat flux (HFSS), and net surface radiation (RNS) for the three CLASSIC ensemble members and for the reference data CLASSr, FluxCom, and FLUXNET. Subplot (b) shows the corresponding fractions with respect to net surface radiation.



**Figure 11.** (a) Zonal mean values and (b) corresponding biases for heat fluxes and variables related to the water cycle.





**Figure 12.** (a) Zonal mean values and (b) corresponding biases for variables related to the carbon cycle.



**Figure 13.** (a) Benchmark scores computed from reference data only, (c) mean model ensemble skill scores, and (b) corresponding score difference.



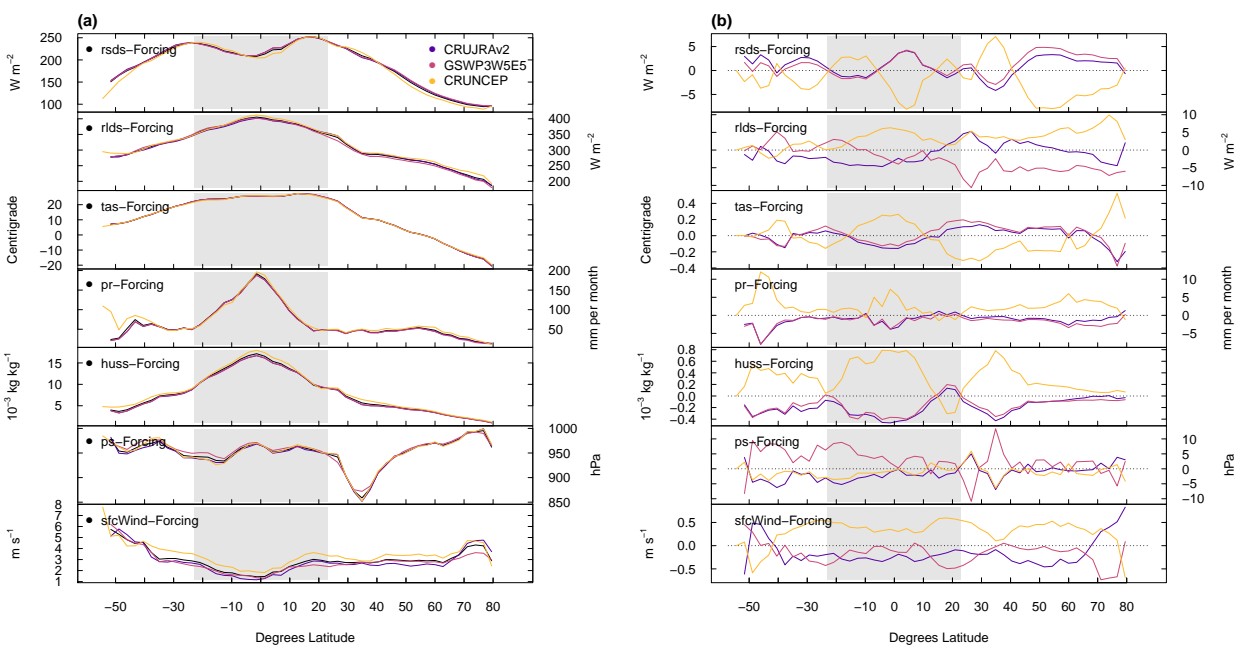

**Figure A1.** (a) Zonal mean values of meteorological forcing data and (b) corresponding difference between the individual forcing and the mean of the three forcings, where *rsds* and *rlds* are the surface downwelling SW and LW radiation, respectively, *tas* is the near-surface air temperature, *pr* is the precipitation, *huss* is the near-surface specific humidity, *ps* is the surface pressure, and *sfcWind* is the near-surface wind speed.





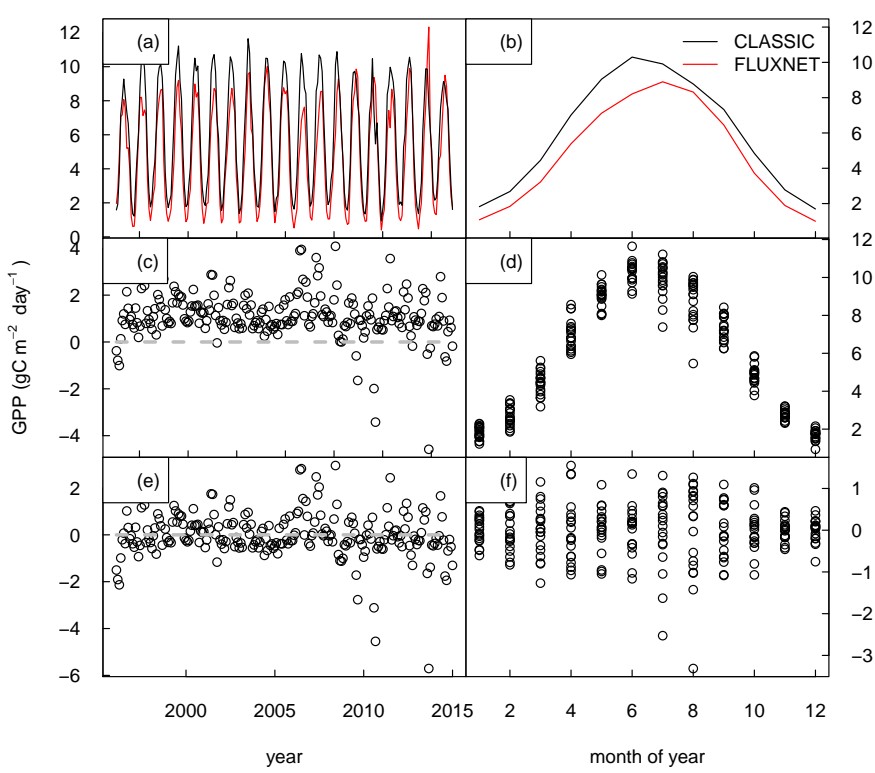

**Figure A2.** The left-hand column shows monthly values of (a) modeled (black) and observed (red) gross primary productivity, (c) residuals, and (e) residuals of the anomalies. The right-hand column gives the corresponding model values for the (b) climatological mean seasonal cycle, (d) monthly variability, and (f) monthly variability of anomalies. The data corresponds to CLASSIC model output for the location of an eddy covariance measurement site in the Netherlands (NL-Loo).



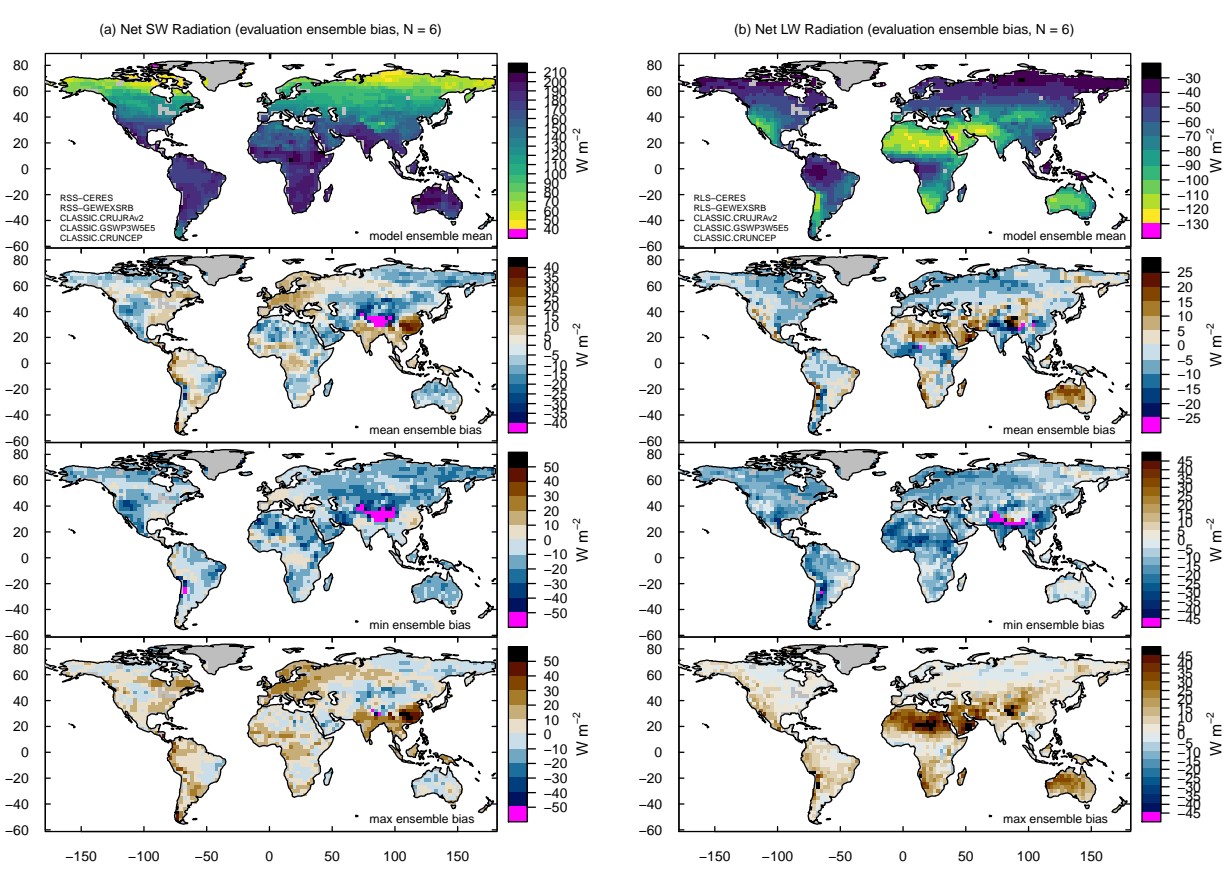

**Figure A3.** Same as Figure 7 but for (a) net SW radiation and (b) net LW radiation.





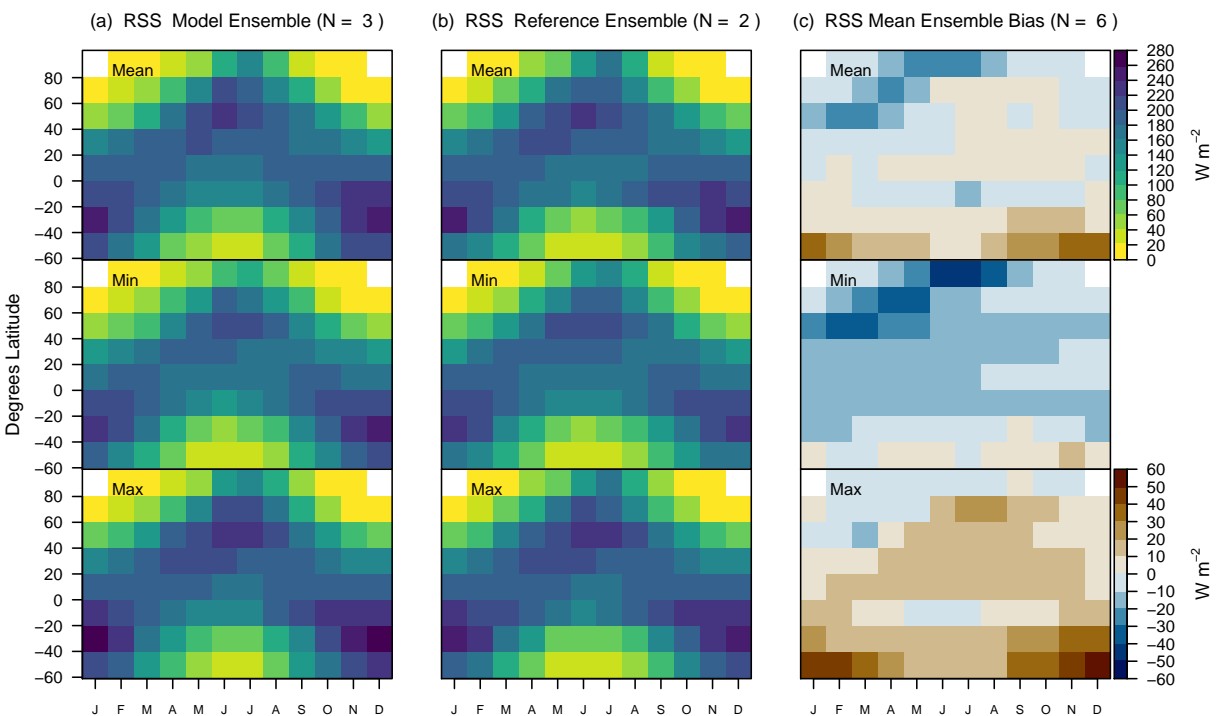

**Figure A4.** Same as in Figure 8 but for net SW radiation.



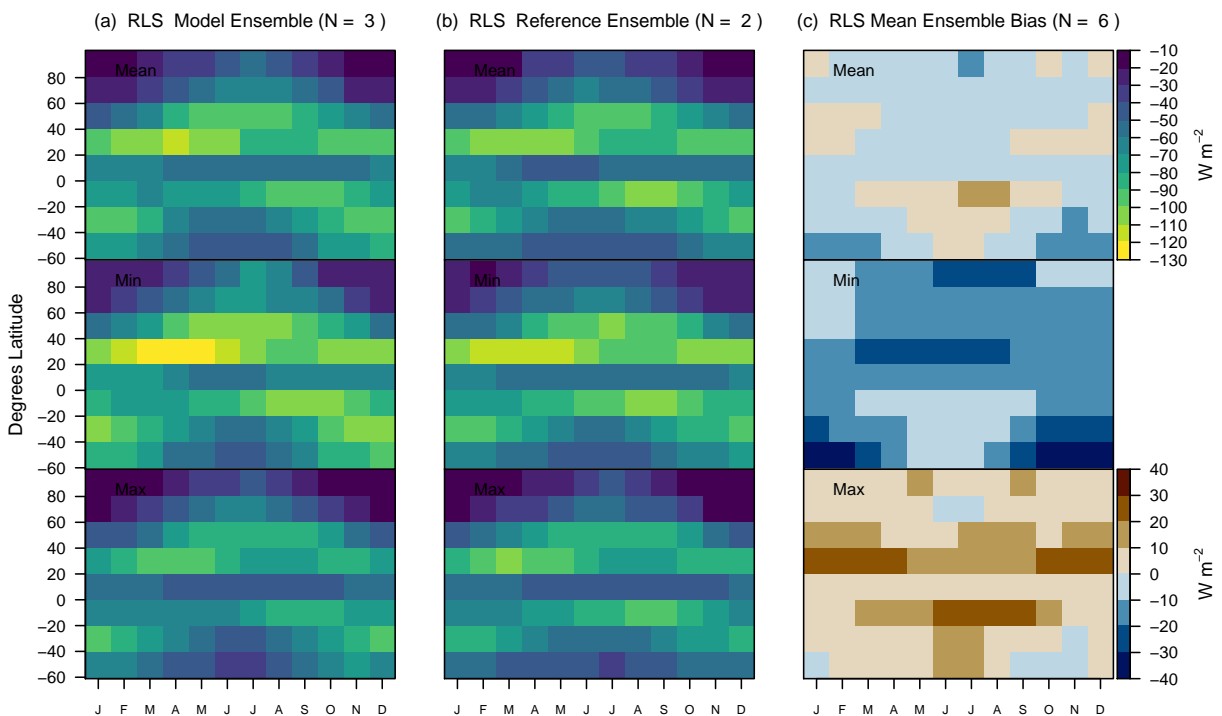

**Figure A5.** Same as in Figure 8 but for net LW radiation.



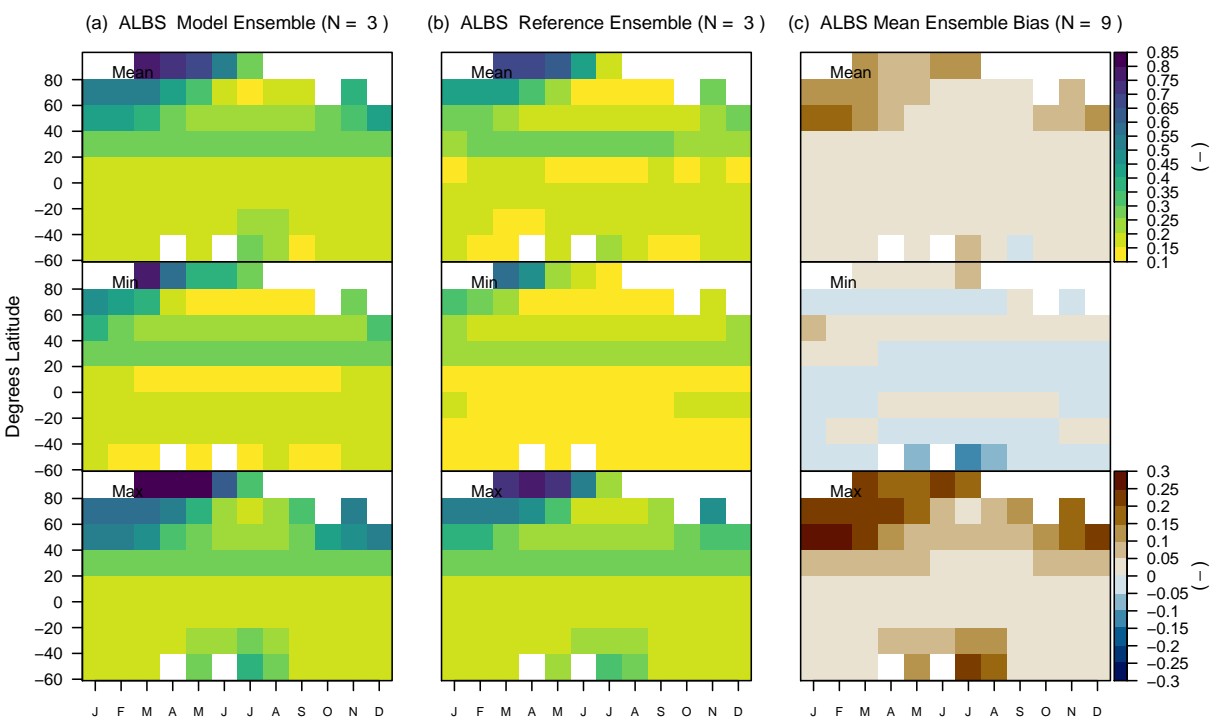

**Figure A6.** Same as in Figure 8 but for surface albedo.



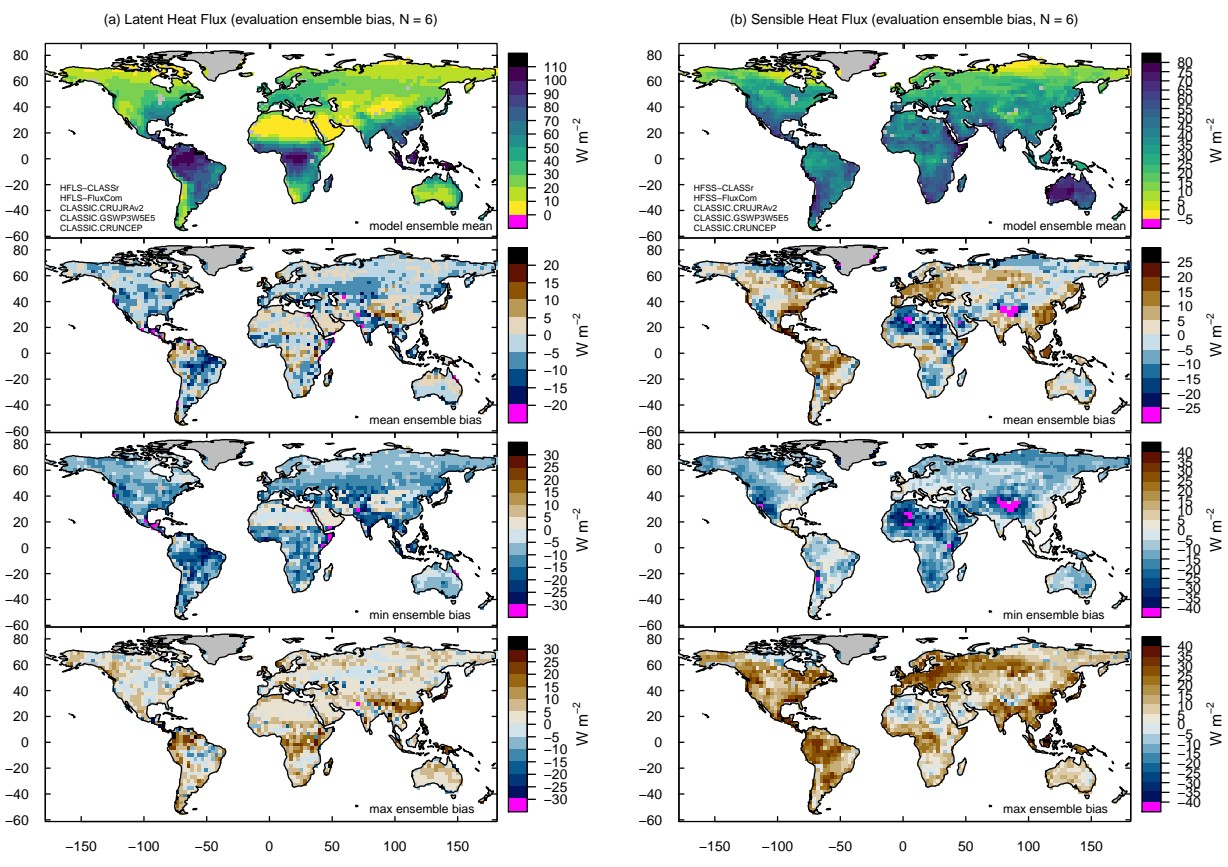

**Figure A7.** Same as Figure 7 but for (a) latent heat flux and (b) sensible heat flux.





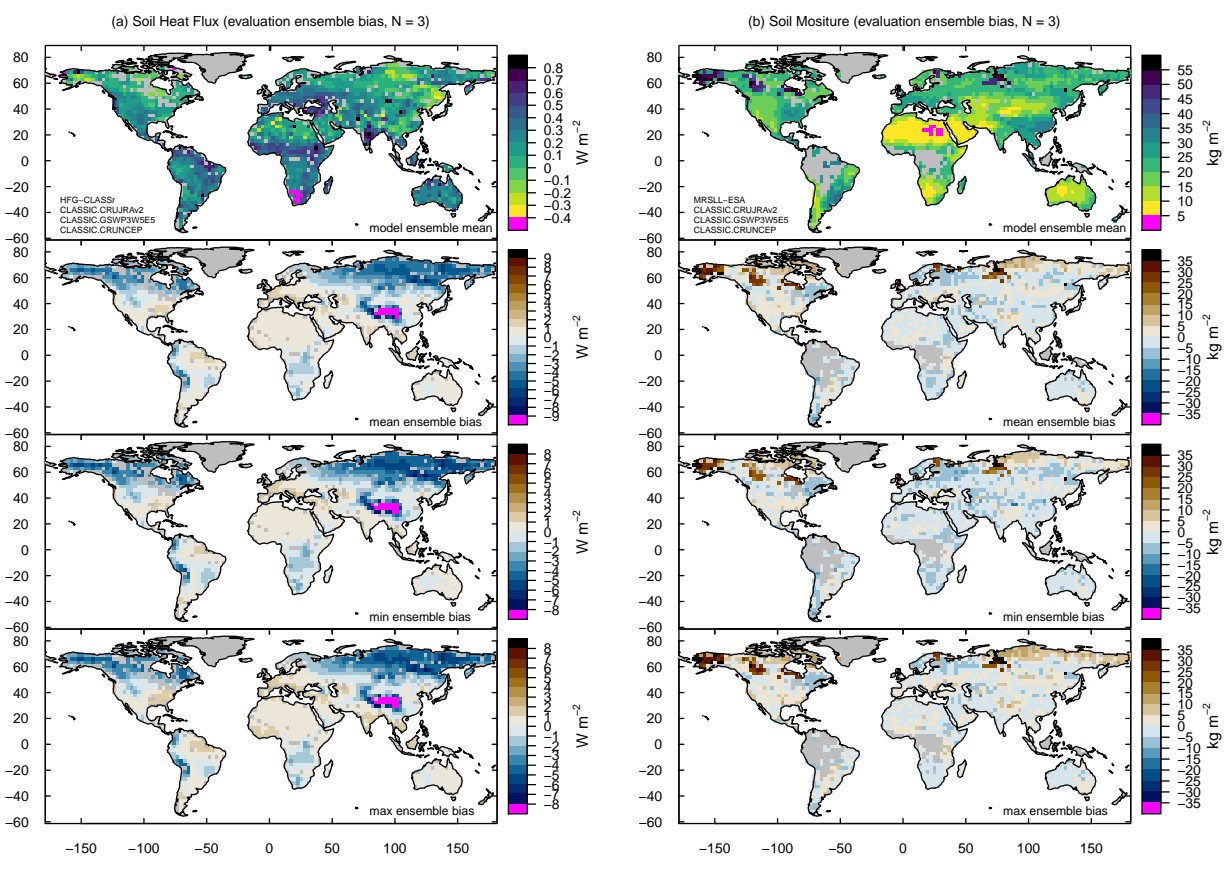

**Figure A8.** Same as Figure 7 but for (a) soil heat flux and (b) soil moisture in the top 10 cm.



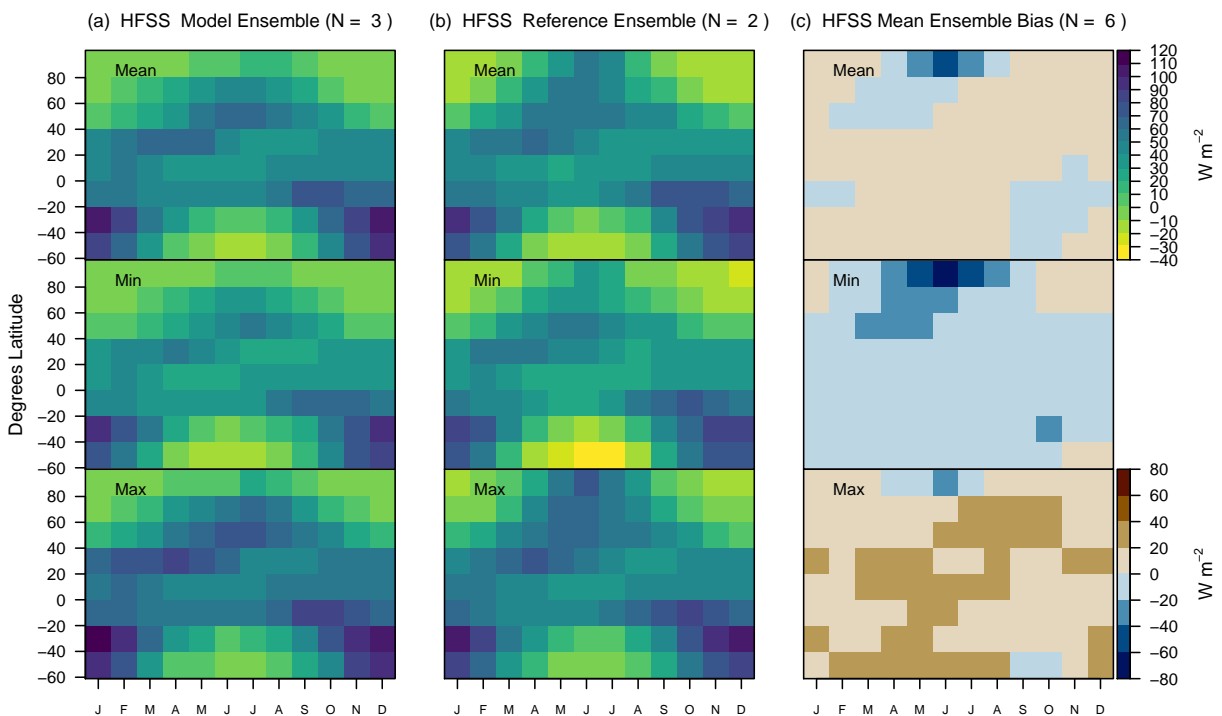

**Figure A9.** Same as in Figure 8 but for sensible heat flux.



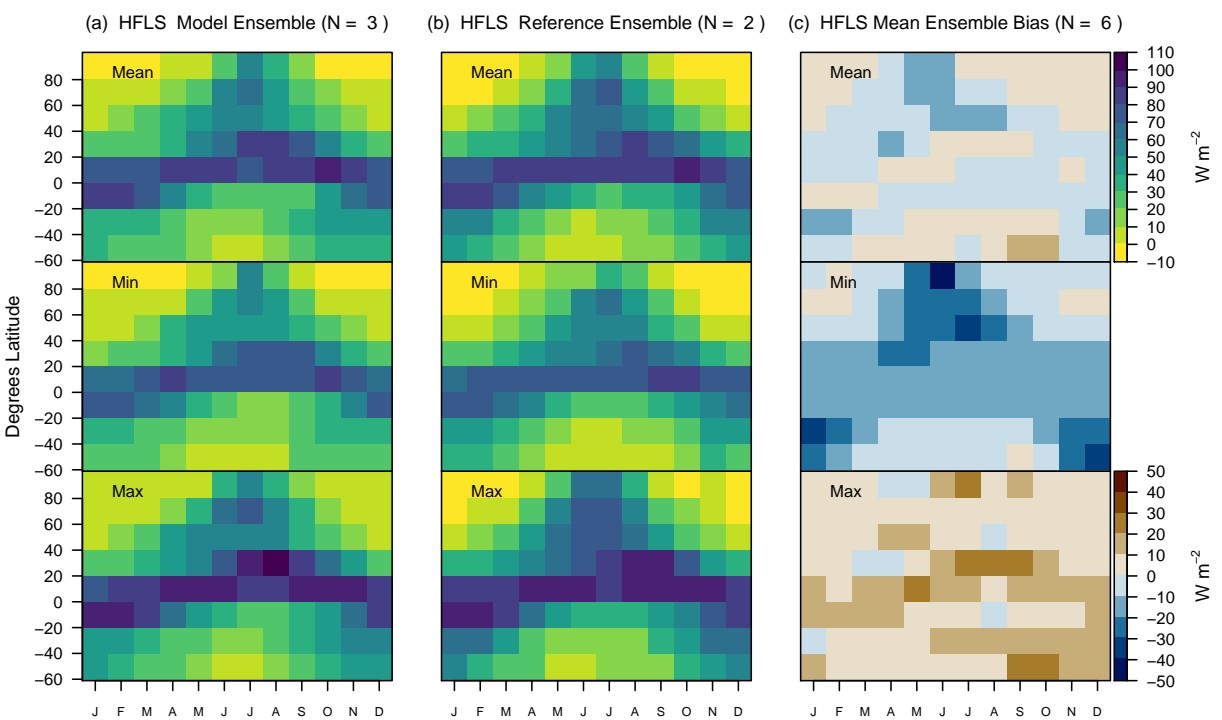

**Figure A10.** Same as in Figure 8 but for latent heat flux.



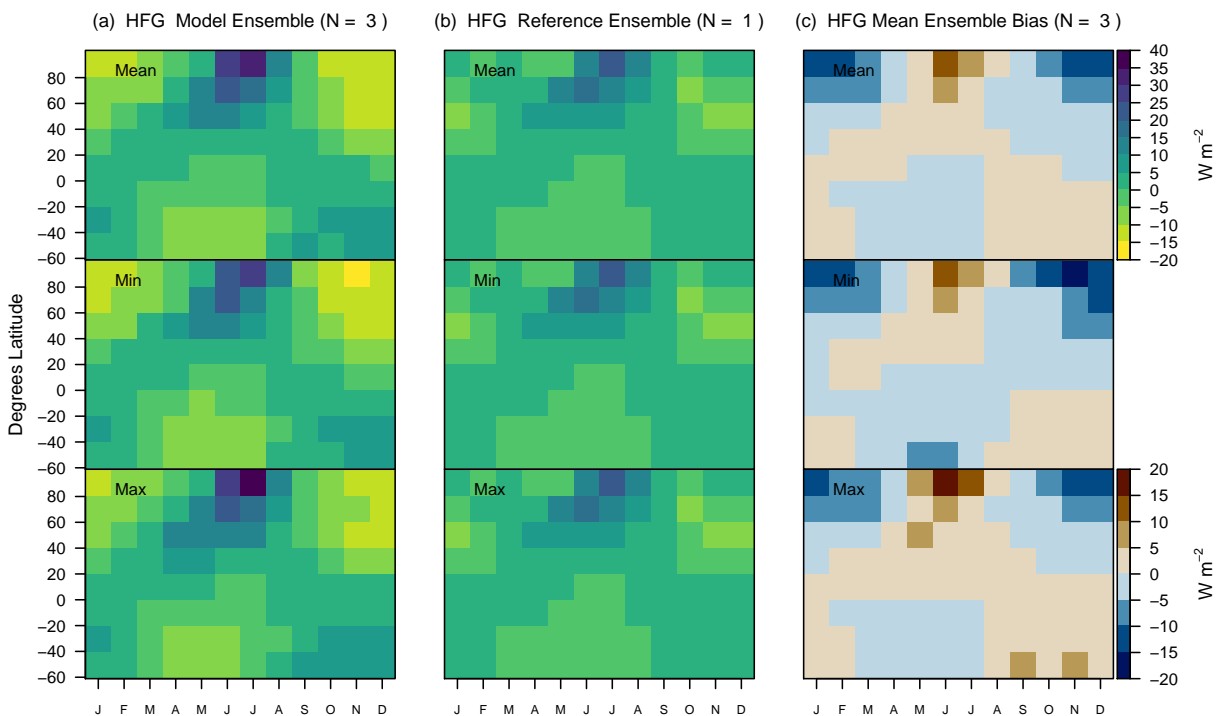

**Figure A11.** Same as in Figure 8 but for soil heat flux.





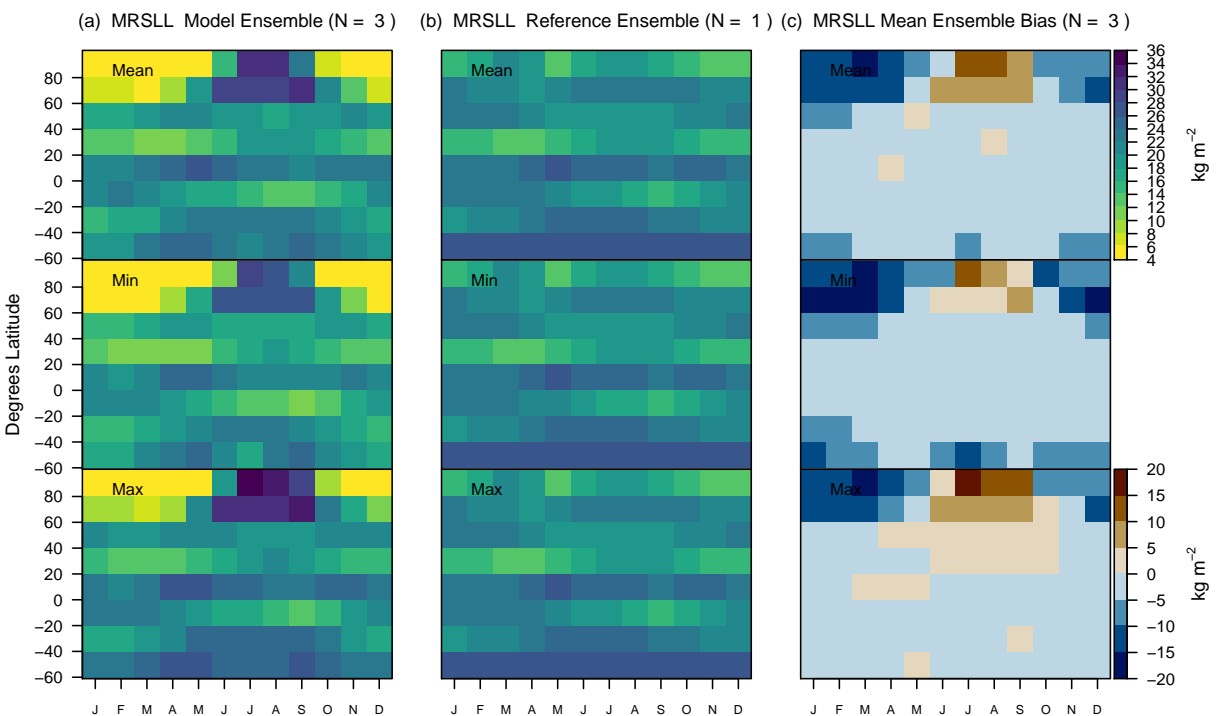

**Figure A12.** Same as in Figure 8 but for soil mositure.



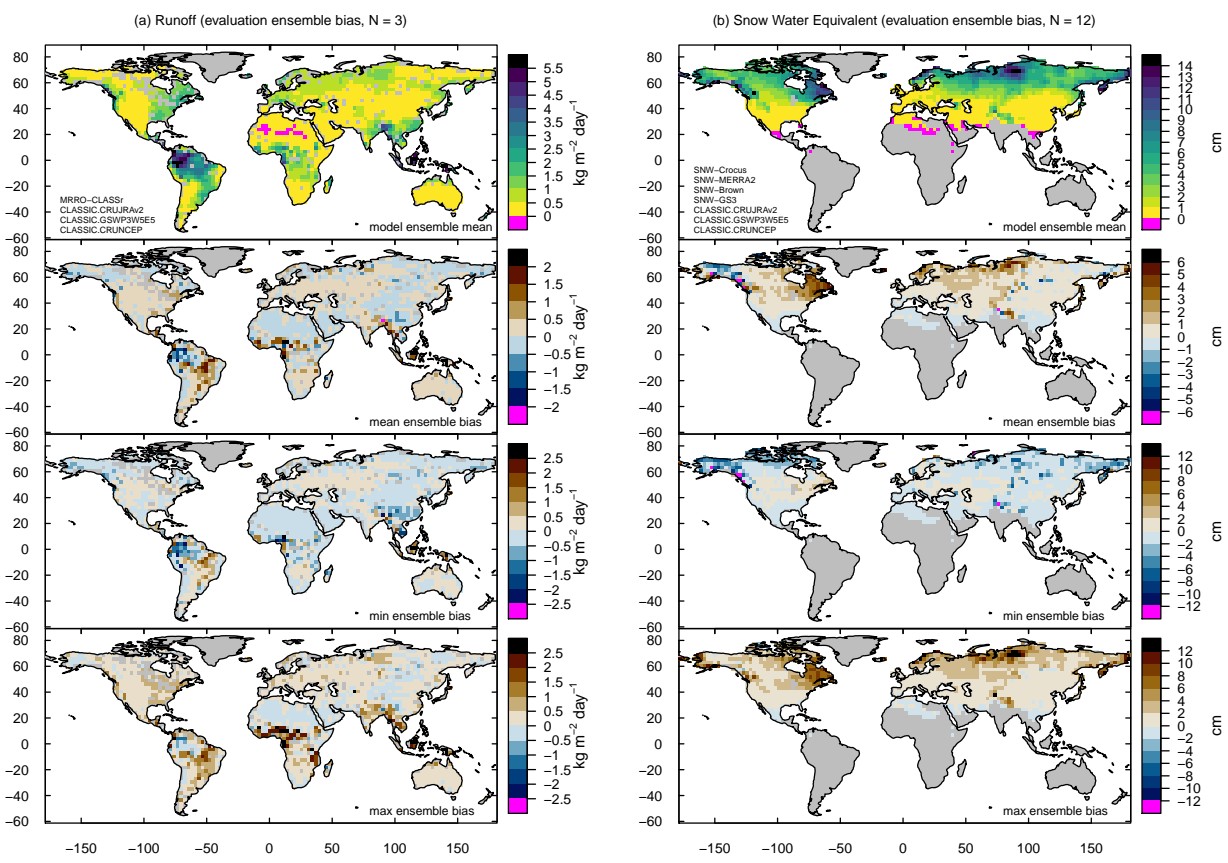

**Figure A13.** Same as Figure 7 but for (a) runoff and (b) snow water equivalent. Magenta-colored grid cells of the model ensemble mean are exactly zero.





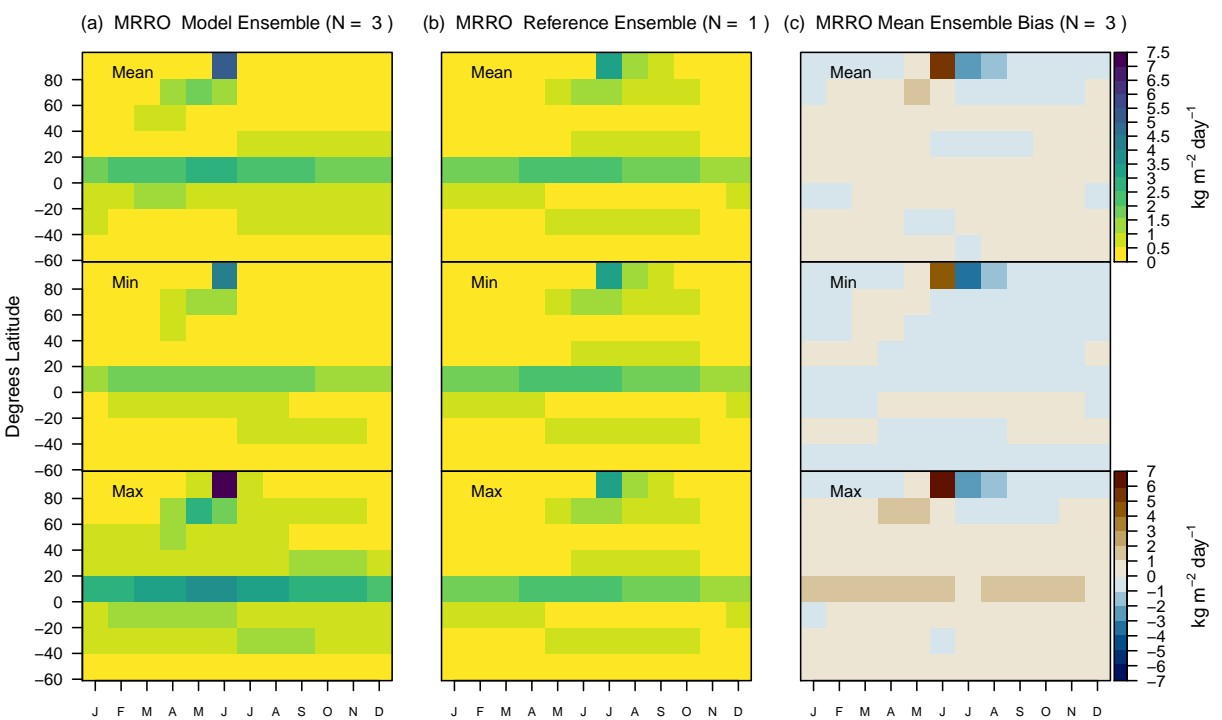

**Figure A14.** Same as in Figure 8 but for runoff.



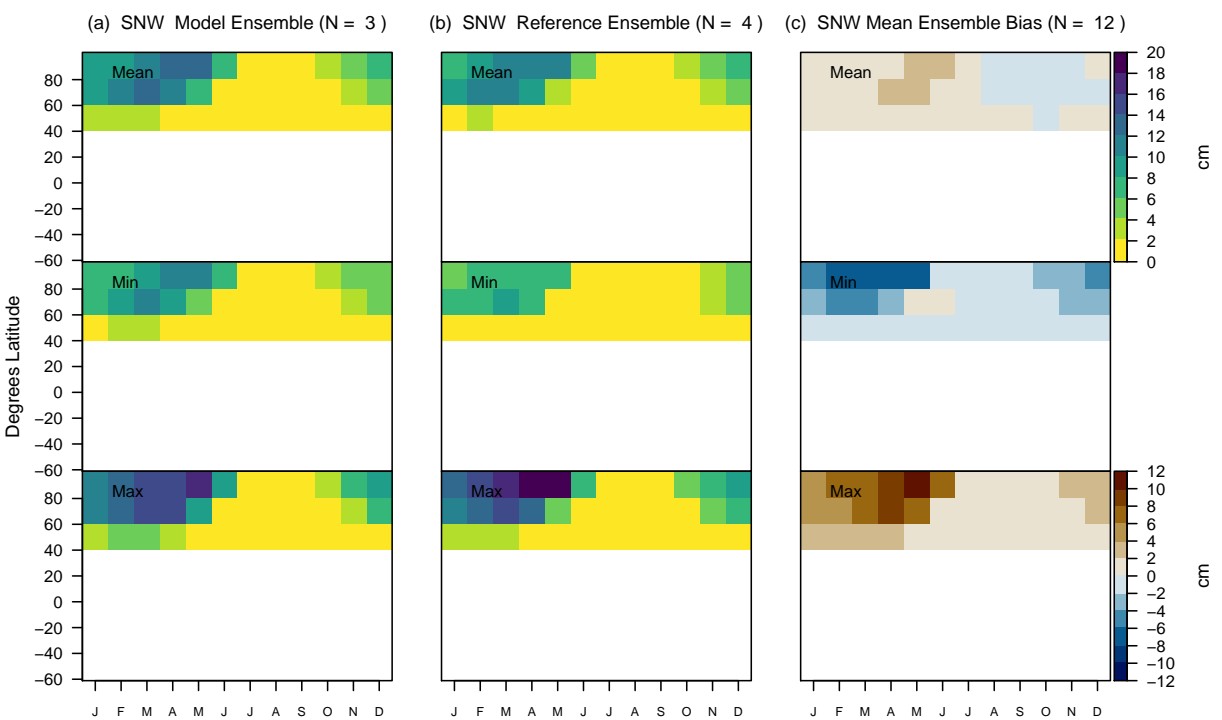

**Figure A15.** Same as in Figure 8 but for snow water equivalent.



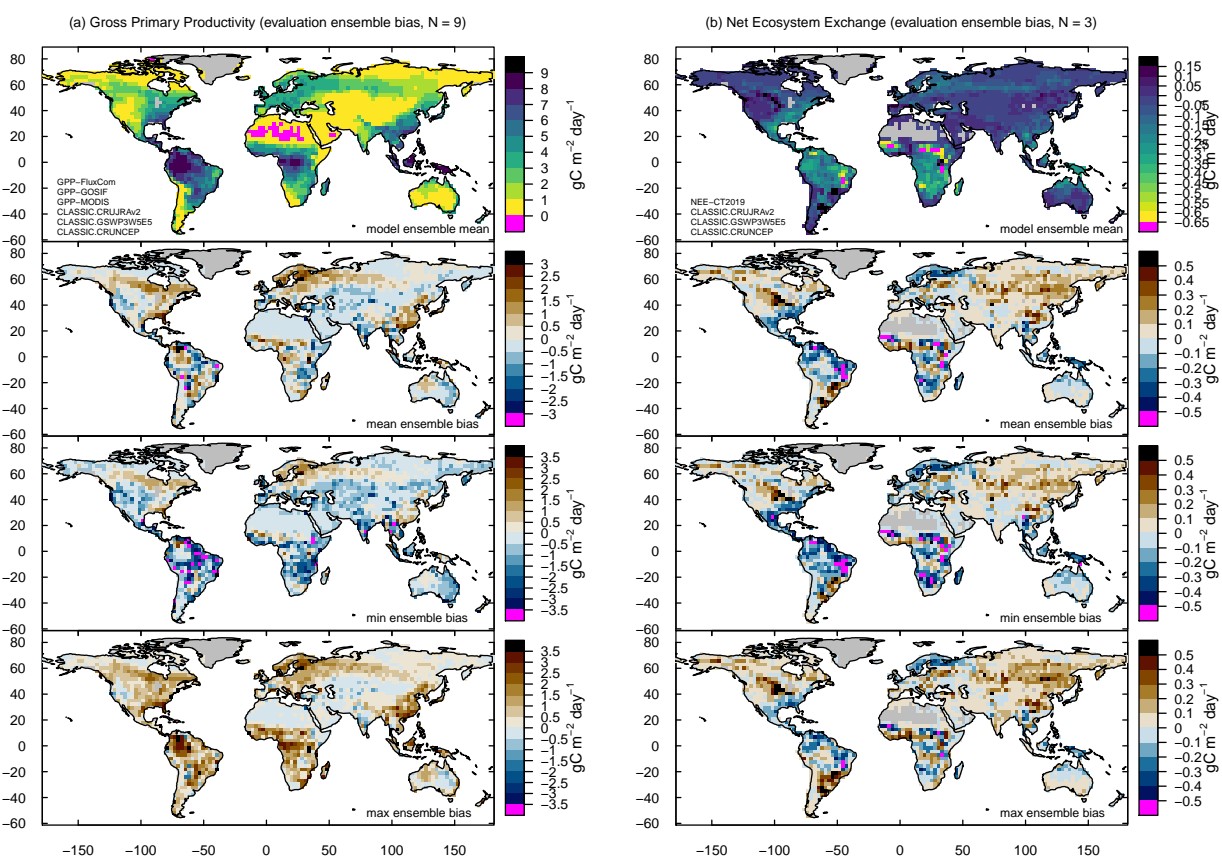

**Figure A16.** Same as Figure 7 but for (a) gross primary productivity and (b) net ecosystem exchange. Magenta-colored grid cells of the model ensemble mean are exactly zero.



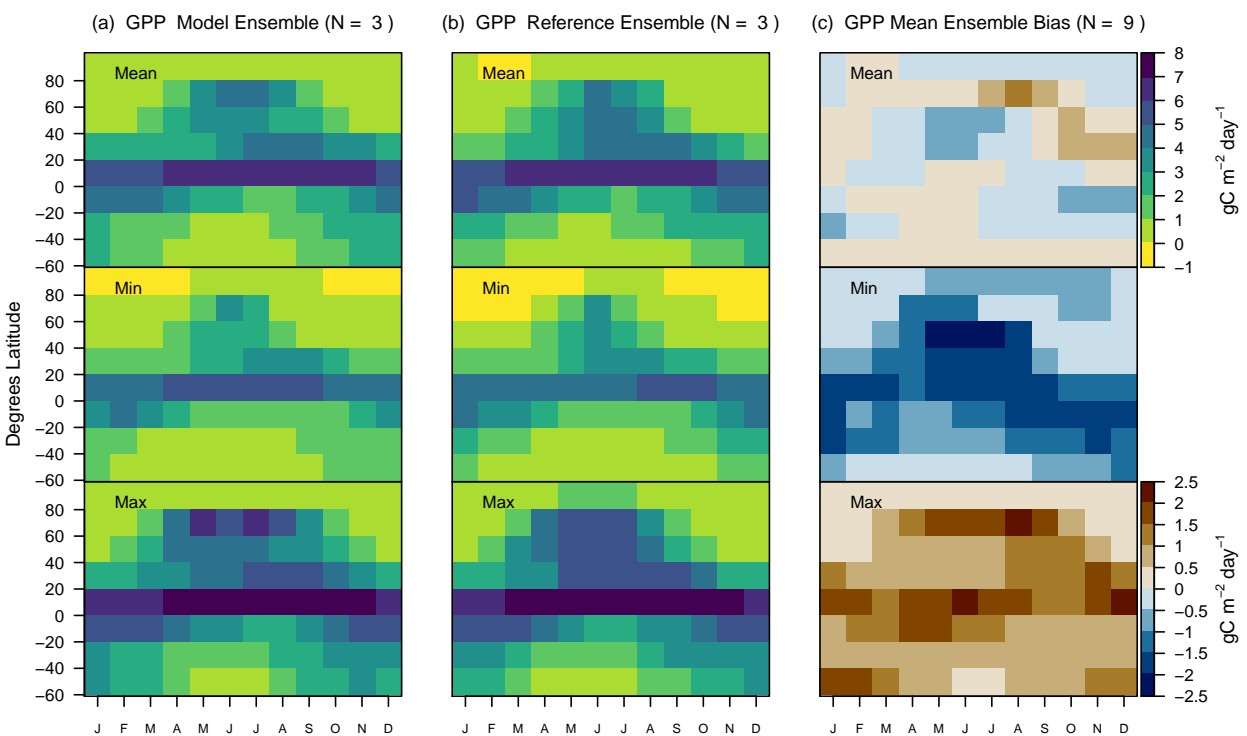

**Figure A17.** Same as in Figure 8 but for gross primary productivity.



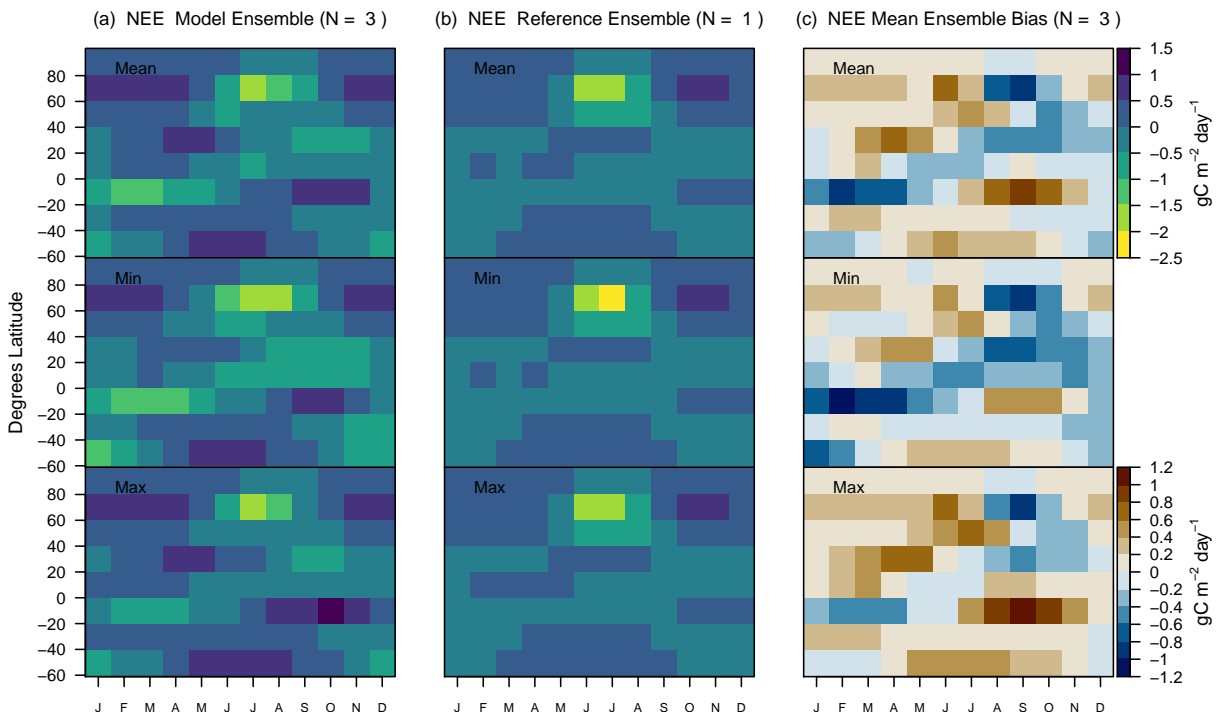

**Figure A18.** Same as in Figure 8 but for net ecosystem exchange.





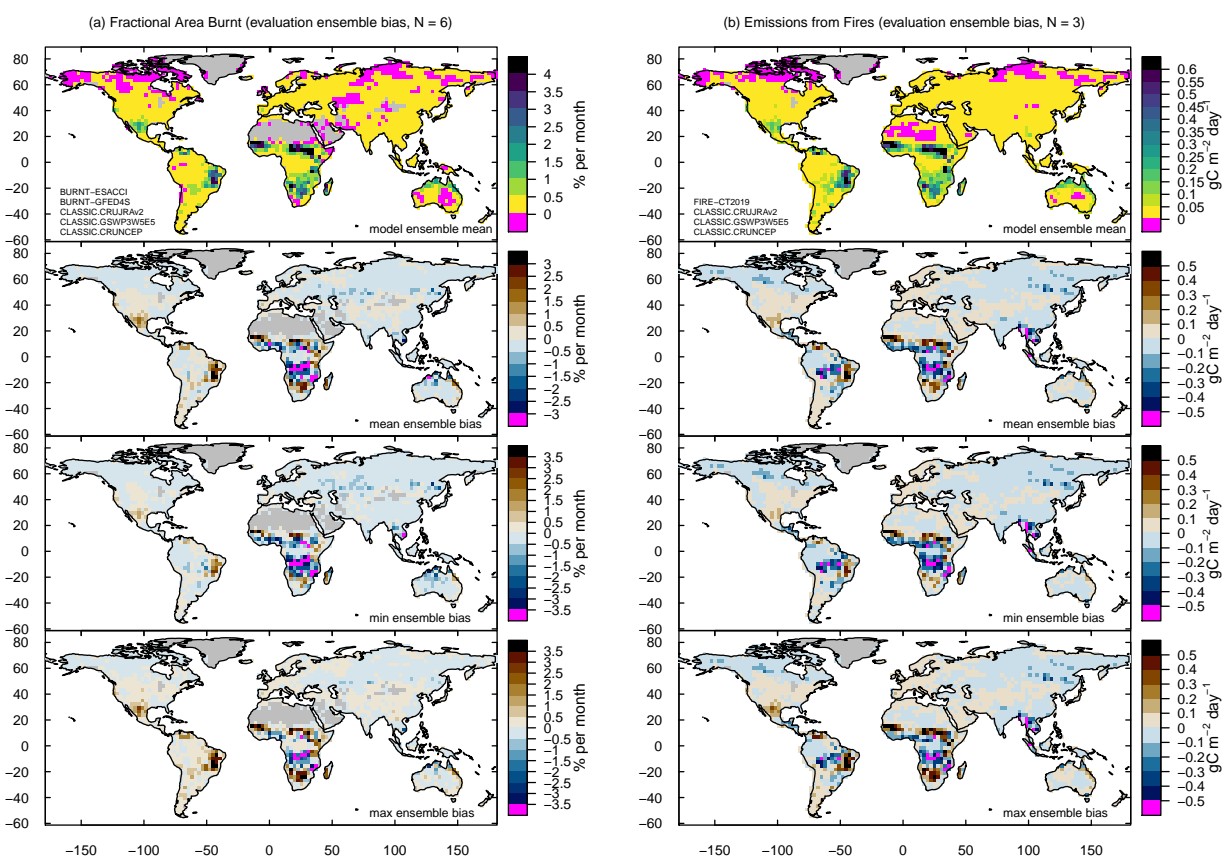

**Figure A19.** Same as Figure 7 but for (a) fractional area burnt and (b) emissions from wild fires. Magenta-colored grid cells of the model ensemble mean are exactly zero.



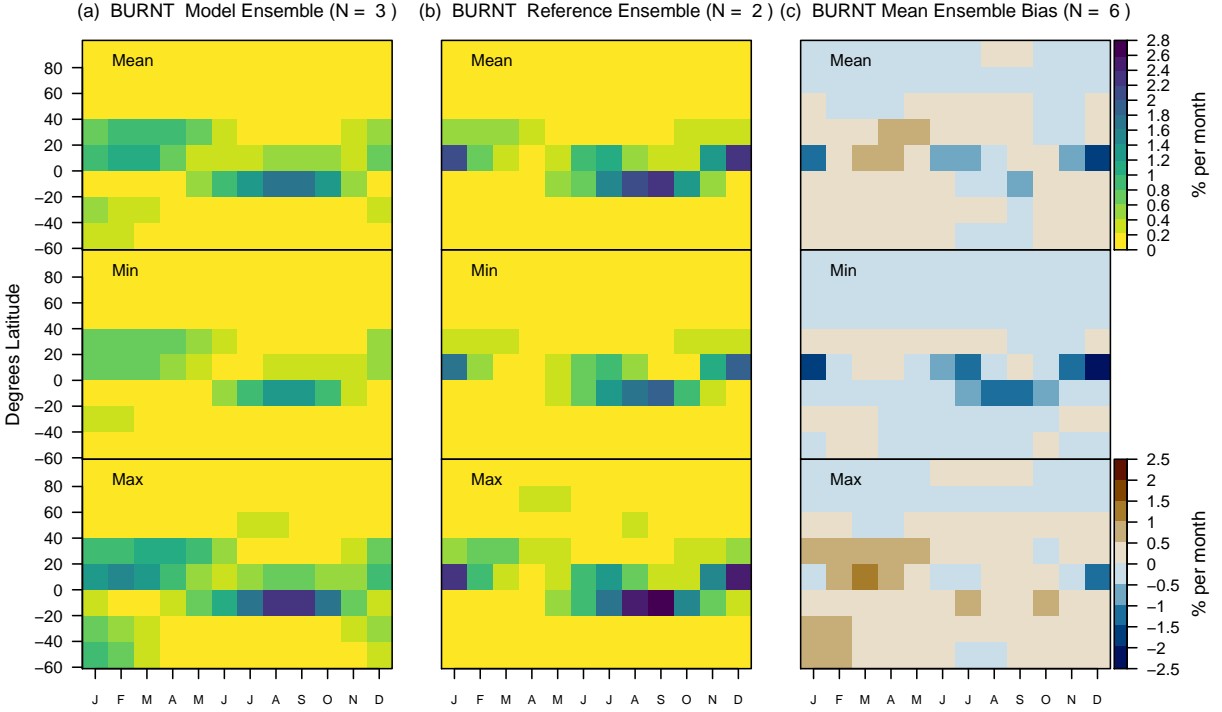

**Figure A20.** Same as in Figure 8 but for fractional area burnt.



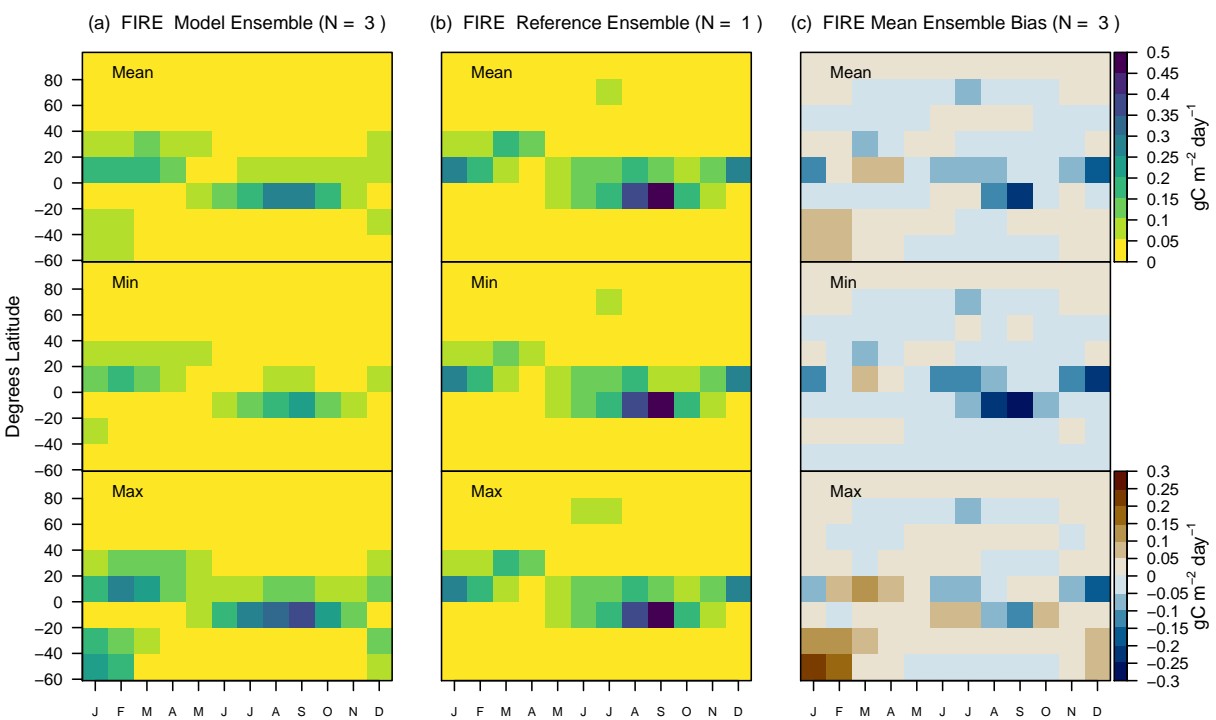

**Figure A21.** Same as in Figure 8 but for emissions from wild fires.



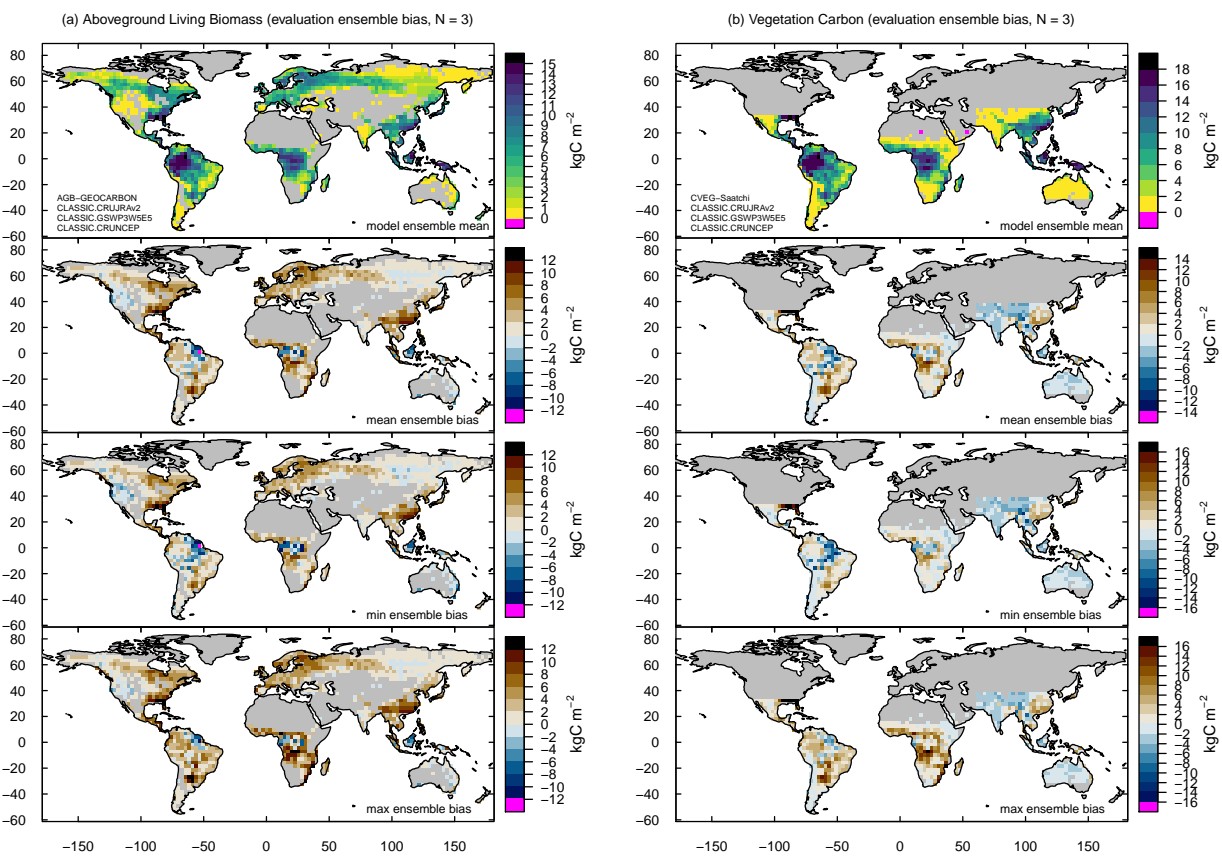

**Figure A22.** Same as Figure 7 but for (a) above-ground living biomass and (b) vegetation carbon in the tropics.



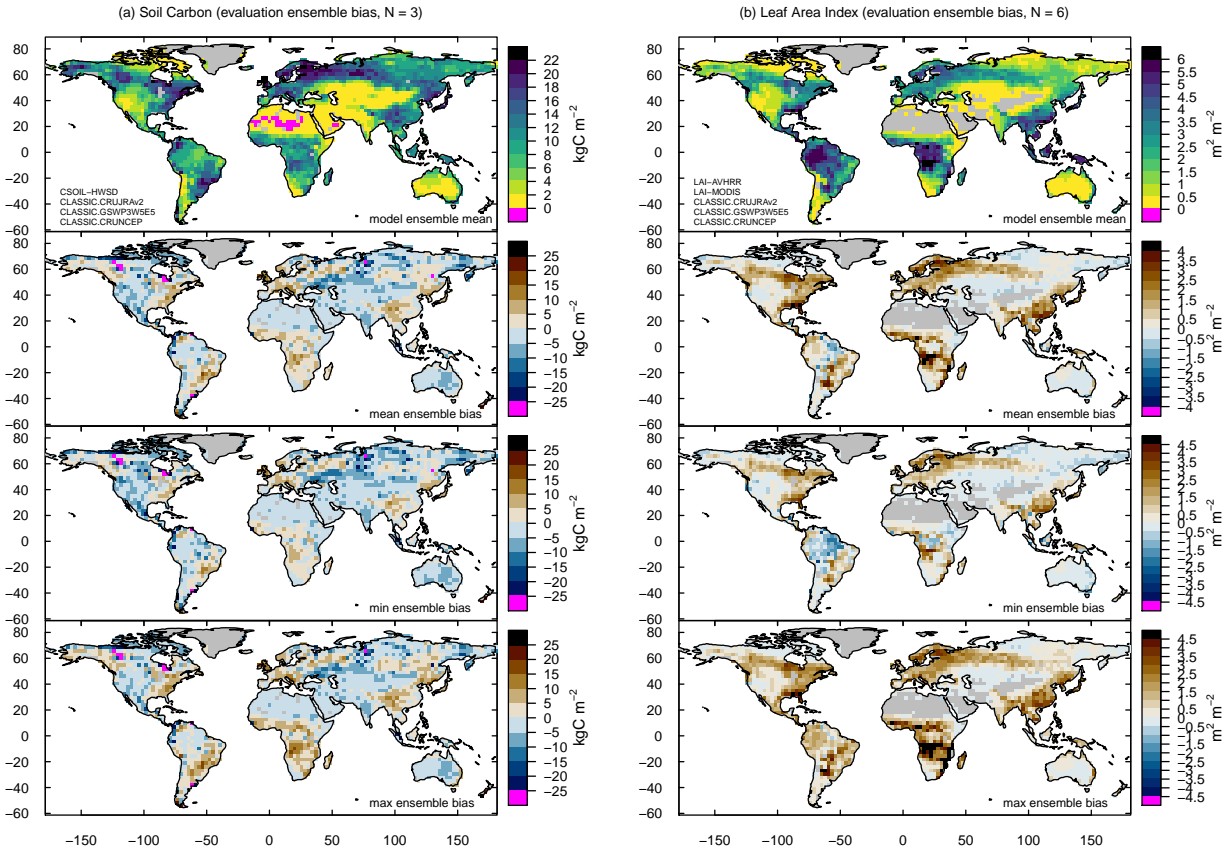

**Figure A23.** Same as Figure 7 but for (a) soil organic carbon mass and (b) leaf area index. Magenta-colored grid cells of the model ensemble mean are exactly zero.





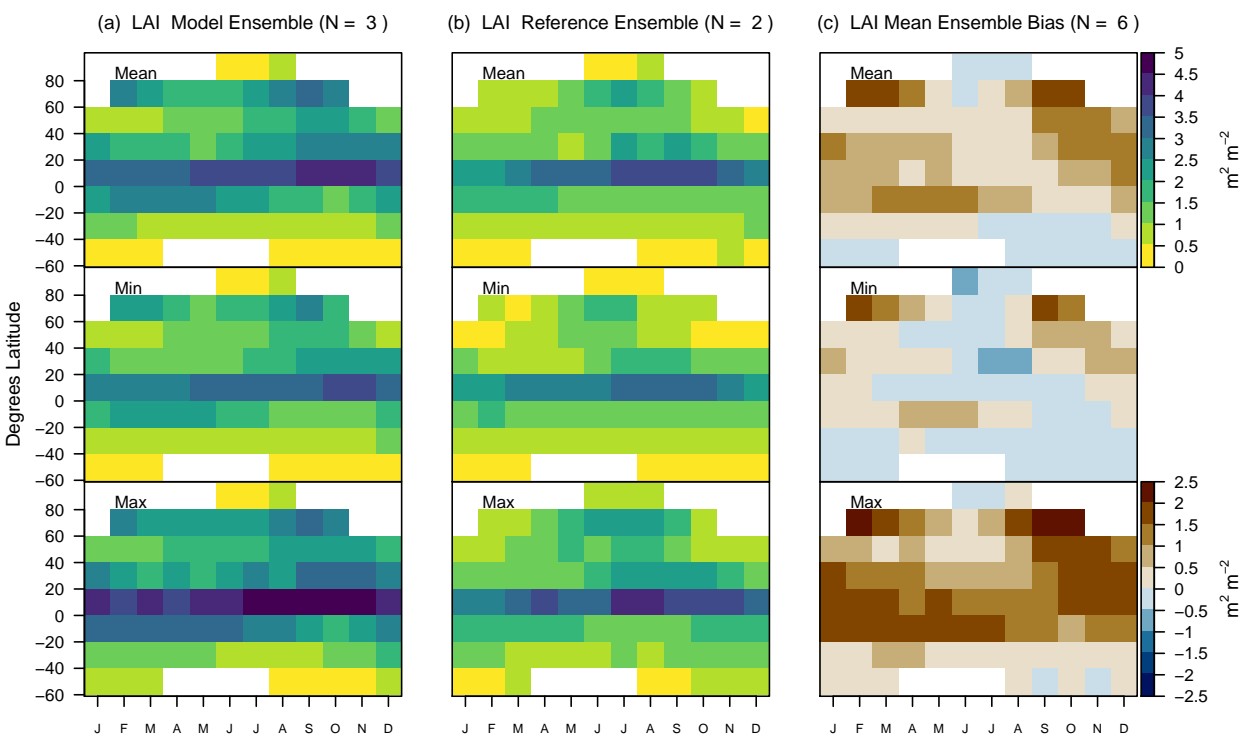

**Figure A24.** Same as in Figure 8 but for leaf area index.





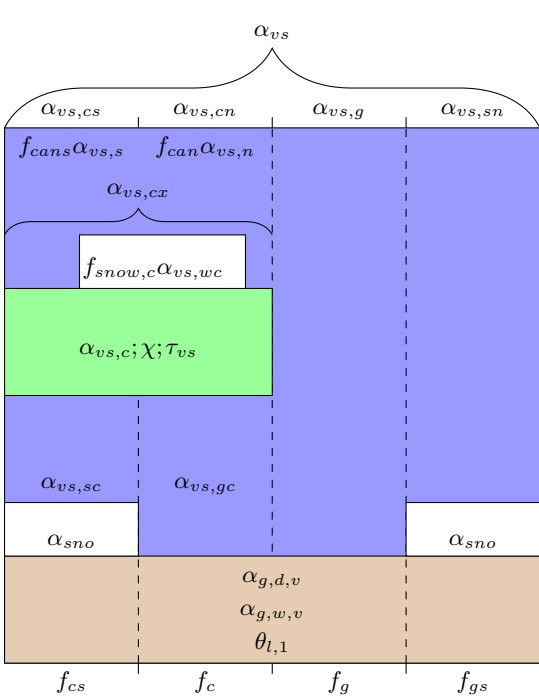

**Figure A25.** Contributions from the bare ground (brown), canopy (green), and snow (white) to the total visible albedo of the land surface. All variables are defined in the Appendix.



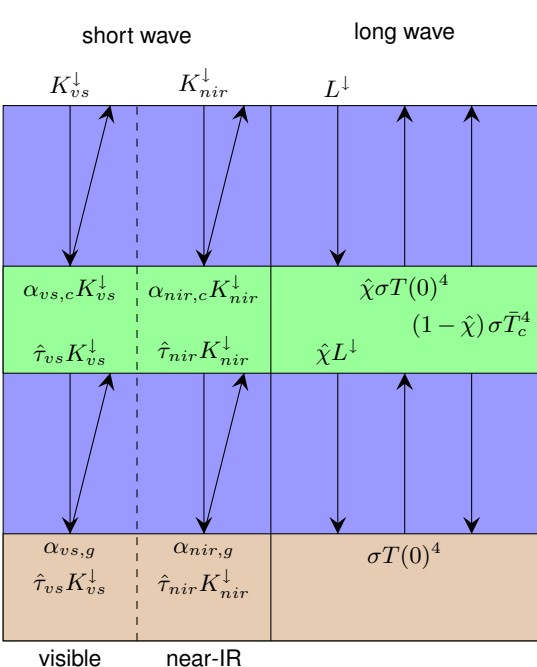

**Figure A26.** Longwave and shortwave radiation fluxes between the ground (brown), canopy (green), and overlying atmosphere. The meaning of each arrow is given at its tail. All variables are defined in the Appendix.