# Peer review of "CLASSIC v1.0: the open-source community successor to the Canadian Land Surface Scheme (CLASS) and the Canadian Terrestrial Ecosystem Model (CTEM) - Part 2: Global Benchmarking"

_Geoscientific Model Development, 2020_

## Referee Comment (RC1) · Anonymous Referee #1 · 16 Dec 2020

This manuscript describes the results of a global benchmarking exercise in which the CLASSIC land surface model is compared with observation-based estimates of various quantities. Uncertainty in driving meteorological data and in the observations is assessed using an ensemble of runs and in the analysis procedure.

I recommend that this manuscript be accepted for publication with minor corrections (it is bit of a toss up between no corrections or minor corrections, but I will go for minor as

there is some small room for improvement).

This is clearly a long manuscript! Despite that I have few comments to make. The nature of the material (model benchmarking) almost inevitably leads to text that is descriptive and repetitive as different aspects/variables/datasets are considered in turn. This level of detail is mainly appreciated by people using the model, or by other modelling groups who wish to understand how models compare, or by someone with a particular interest in one sub-area (say energy fluxes) – but in general it does not make for a paper that is particularly easy to read or to review. Thus I am quite happy to admit that I have not studied every line in every table, nor every figure. However, I am reassured by the quality of what I have seen and am fairly confident that the many details provided would also prove satisfactory (if I had the time and inclination to study them). This might sound like slightly veiled criticism but is intended as an endorsement of the manuscript!

I particularly value the treatment of uncertainty, with three meteorological datasets used to drive the model and more than one "observation-based" dataset used to assess most of the variables. This should really be standard practice but often lack of time encourages modellers to consider a reduced set of possibilities. The value of the approach taken here is shown by the headline result that for 10 of 19 variables the sign of the bias depends on the datasets used. Narrowing these uncertainties is a key challenge for the land surface community, but beyond the scope of this paper.

In general I think this is an excellent attempt to benchmark (or evaluate; some people distinguish these terms) a land surface model, with a commendable level of detail presented (without being completely overwhelming). I think some other modelling centres will look at this with some envy!

**Specific points**

Is it possible to make more of the comparison with the results of Part 1 (Melton et al.)? At present I think this is limited to the brief observation in the Discussion that the "overall

statistic" is similar. Given that a holy grail for this kind of work is to get a coherent picture of model performance across multiple variables/sites/scales/datasets, it might be good to pursue the comparison to the results of Melton et al. in slightly greater depth (though I appreciate that it might be difficult to draw insightful conclusions beyond the fact that the results are similar).

Similarly, there is little or no mention of benchmarking studies using other models (e.g. CLM) and any common conclusions. Again, this might be difficult, but some brief mention would be warranted.

Are all simulations carried out at the same spatial resolution and/or at the resolution of each meteorological dataset? I suspect all the datasets are 0.5deg.

Figures 1 & 2 (and some others, such as Fig.7) - the colour scale could be better, in particular to distinguish the darker tones (e.g. >5 and <-40 in Fig.1a). This is particularly important for Figs. 1&2 because of the small size of the symbols - they pretty much all look the same dark colour to me. Can the symbols be made larger and/or the panels larger? Without that these plots are of very limited value.

Line 214: "it's" should be "its"

There is repeated but not ubiquitous use of italics when numbering lists (e.g. $i$, $ii$)- which looks a bit odd.

Fig.A1 - Eyeballing this it looks like CRUNCEP is often rather anomalous compared to the other two datasets (e.g. SW, Tas, precip, huss, wind). Is that because the other products are based on more similar data sources, and/or might this suggest that CRUNCEP is an anomalous (possibly inferior) dataset?

---

## Author Comment (AC1) · 9 Feb 2021

**Anonymous Referee (1)**

**RC1:**This manuscript describes the results of a global benchmarking exercise in which the CLASSIC land surface model is compared with observation-based estimates of various quantities. Uncertainty in driving meteorological data and in the observations is assessed using an ensemble of runs and in the analysis procedure. I recommend that this manuscript be accepted for publication with minor corrections (it is bit of a toss up between no corrections or minor corrections, but I will go for minor as there is some small room for improvement). This is clearly a long manuscript! Despite that I have few comments to make. The nature of the material (model benchmarking) almost inevitably leads to text that is descriptive and repetitive as different aspects/variables/datasets are considered in turn. This level of detail is mainly appreciated by people using the model, or by other modelling groups who wish to understand how models compare, or by someone with a particular interest in one sub-area (say energy fluxes) - but in general it does not make for a paper that is particularly easy to read or to review. Thus I am quite happy to admit that I have not studied every line in every table, nor every figure. However, I am reassured by the quality of what I have seen and am fairly confident that the many details provided would also prove satisfactory (if I had the time and inclination to study them). This might sound like slightly veiled criticism but is intended as an endorsement of the manuscript!

I particularly value the treatment of uncertainty, with three meteorological datasets used to drive the model and more than one "observation-based" dataset used to assess most of the variables. This should really be standard practice but often lack of time encourages modellers to consider a reduced set of possibilities. The value of the approach taken here is shown by the headline result that for 10 of 19 variables the sign of the bias depends on the datasets used. Narrowing these uncertainties is a key challenge for the land surface community, but beyond the scope of this paper. In general I think this is an excellent attempt to benchmark (or evaluate; some people distinguish these terms) a land surface model, with a commendable level of detail presented (without being completely overwhelming). I think some other modelling centres will look at this with some envy!

Reply: Thank you for your positive evaluation of our manuscript and the very encouraging remarks!

Specific points

- **RC1:** Is it possible to make more of the comparison with the results of Part 1 (Melton et al.)? At present I think this is limited to the brief observation in the Discussion that the "overall statistic" is similar. Given that a holy grail for this kind of work is to get a coherent picture of model performance across multiple variables/sites/scales/datasets, it might be good to pursue the comparison to the results of Melton et al. in slightly greater depth (though I appreciate that it might be difficult to draw insightful conclusions beyond the fact that the results are similar).

  Reply: We now compare the overall score values between both studies in the Discussion section. Given that results are similar, we conclude that an evaluation of global model outputs against eddy covariance data is meaningful, despite the fact that the present study does not use site-specific model inputs. We emphasize that this is particularly important for interpreting the low scores for NEE, which cannot be explained by using non-site specific forcing. We have now added two paragraphs in the Introduction and Discussion section.

In the Introduction section: "A first evaluation of CLASSIC has been presented by Melton et al. (2020), who ran the model for 31 FLUXNET sites using locally observed meteorological data. Model outputs were compared against observed net surface radiation (RNS), latent heat flux (HFLS), sensible heat flux (HFSS), GPP, ecosystem respiration (RECO), and NEE. Results showed that CLASSIC reproduced the RNS, HFLS, HFSS, GPP, and RECO observations reasonably well. Reproducing NEE fluxes proved to be more challenging. While for some sights modeled NEE fluxes were in good agreement with measurements (e.g. site FI-Hyy, Figure S22 and S23 in Melton et al. 2020), for most sites NEE differed considerably from observations. Assessing the annual mean NEE value for each station showed that modeled and observed annual mean values were not correlated across stations. The poor model performance for NEE may be related to the fact that the disturbance history of the site is not reflected in the model. A comparison between results presented in the present study and Melton et al. (2020) is provided in the Discussion section."

In the Discussion section: "The overall score ($S_{overall}$) values documented in this study are similar to the ones reported by Melton et al. (2020). The values are 0.66 (0.69) for GPP, 0.71 (0.66) for HFLS, 0.68 (0.66) for HFSS, 0.47 (0.44) for NEE, 0.64 (0.65) for RECO, and 0.8 (0.82) for RNS, where values in parenthesis correspond to results from Melton et al. (2020). The fact that both approaches yield similar results suggests that a comparison of global model output and eddy covariance data at the grid cell level provides meaningful results. This finding implies that the low scores for NEE documented here are not due to the fact that we used global rather than site-specific model inputs. However, it must be noted that interpreting the differences between both studies is challenging, given that our evaluation is based on 204 sites, while Melton et al. (2020) used 31 sites. Furthermore, our approach averages observations across sites that are located within the same model grid cell and cover the same time period."

- **RC1:** Similarly, there is little or no mention of benchmarking studies using other models (e.g. CLM) and any common conclusions. Again, this might be difficult, but some brief mention would be warranted.

  Reply: We now included the key findings by Lawrence et al. (2019) and Bonan et al. (2019) who used ILAMB to benchmark CLM using three different model versions and three different meteorological forcing data sets.

  In the Introduction section: "Lawrence et al. (2019) applied ILAMB to quantify how model development affected model performance for the Community Land Model (CLM). While the authors found a general improvement from CLM version 4 to version 5, they also identified variables where model performance would improve or worsen, depending on the choice of reference data. A second source of uncertainty identified by the authors was related to the choice of meteorological forcing. Driving CLM with three different data sets (GSWP3, CRUNCEPv7, and WATCH/WFDEI) showed that model performance was generally better when forcing CLM with the Global Soil Wetness Project forcing data set (GSWP3), and that the uncertainty associated with the forcing is too large to be neglected (Bonan et al. 2019)."

  In the Discussion section, we now write: "Our results show that model performance is very sensitive to the choice of meteorological forcing and reference data, confirming recent findings presented in Lawrence et al. (2019) and Bonan et al. (2019)."

  Also, please note that we do discuss benchmarking results for all land surface models that participated in TRENDY in the Introduction and Discussion section.

- **RC1:** Are all simulations carried out at the same spatial resolution and/or at the resolution of each meteorological dataset? I suspect all the datasets are 0.5deg.

  Reply: The section *Reference data* mentions that "All gridded reference data are

spatially interpolated to the spatial resolution of a CLASSIC gridcell (2.8125° × 2.8125°)."

- **RC1:** Figures 1 & 2 (and some others, such as Fig.7) - the colour scale could be better, in particular to distinguish the darker tones (e.g. >5 and <-40 in Fig.1a). This is particularly important for Figs. 1 & 2 because of the small size of the symbols - they pretty much all look the same dark colour to me. Can the symbols be made larger and/or the panels larger? Without that these plots are of very limited value.

  Reply: We have now scaled the size of the symbols in proportion to the size of the bias, where larger biases lead to larger symbols. To avoid that the larger symbols hide smaller symbols, the order of plotting is such that smaller symbols are plotted on top of larger symbols. Furthermore, we changed the colors of the extremes, and replaced the "vik" color scheme with the "roma" color scheme (Crameri et al. 2020). While both color schemes are color-blind friendly, the latter seems better for distinguishing darker tones. Please find an example plot below.

- **RC1:** Line 214: "it's" should be "its"

  Reply: Thank you, done.

- **RC1:** There is repeated but not ubiquitous use of italics when numbering lists (e.g. i, ii)- which looks a bit odd.

  Reply: Thank you, done.

- **RC1:** Fig.A1 - Eyeballing this it looks like CRUNCEP is often rather anomalous compared to the other two datasets (e.g. SW, Tas, precip, huss, wind). Is that

because the other products are based on more similar data sources, and/or might this suggest that CRUNCEP is an anomalous (possibly inferior) dataset?

Reply: Yes, with the exception of surface pressure, CRUNCEP differs most, while CRUJRAv2 and GSWP3W5E5 are more similar. This finding was counterintuitive, as CRUNCEP and CRUJRAv2 both rely on CRU. Understanding why these data sets differ and identifying which provides more accurate values is beyond the scope of this manuscript. We have now added the following sentence in the *Meteorological forcing* section:

"With the exception of surface pressure, the data from CRUJRAv2 and GSWP3W5E5 tend to be more similar compared to CRUNCEP."

In the *Conclusion section* we added that:

"Future research could explore the question why meteorological forcing data sets used in this study differ and how this uncertainty could be reduced.".

**References**

Crameri, Fabio, Grace E. Shephard, and Philip J. Heron. 2020. "The Misuse of Colour in Science Communication." Nature Communications 11 (1): 5444.

[Figure]

(a) Net surface radiation
Bias RNS CLASSIC.CRUJRAv2 vs FLUXNET from 1991−01 to 2014−12

**Fig. 1.**

---

## Referee Comment (RC2) · Anonymous Referee #2 · 26 Feb 2021

Overall assessment:

The manuscript submitted by Christian Seiler and coworkers present the evaluation of the open-source community land-surface model CLASSIC v1.0. A wide range of variables related to energy, water and carbon cycle are compared against observation-based, either site- or global-gridded, data.

An extensive evaluation and huge work have been carried out and are presented here

with a high degree of clarity. The manuscript is very well organized and written, and a large and useful selection of figures and tables support this work. This manuscript is of strong interest, not only as an overview of the strengths and weaknesses of one specific model, CLASSIC, but also as a guide for the long but necessary land-surface model evaluation exercice. I detail here minor corrections and feedbacks to be considered, and I warmly recommend this manuscript for publication in GMD.

Corrections and feedbacks:

In the section 2.1 presenting the model tools, CLASSIC is among others presented as a dynamic vegetation model. However, in this work and generally, it is not clear to me if the vegetation distribution is indeed calculated "dynamically" by the model, depending in particular on temperature and CO2 conditions, or prescribed based on land map forcings. Could the authors clarify this point in the manuscript ? For a generally understanding as well: is nitrogen cycle included in this model ?

Correction page 4, line 114: a "c" in missing in "The protcol consists of a spin up".

Correction page 17, line 493: replace "most" by "more" in "the positive biases are most evident in the NH extratropics rather than in the tropics".

The authors underline both in the Abstract and in the Conclusion that "Our results will serve as a baseline for guiding and monitoring future CLASSIC development." Regarding the development monitoring, this manuscript is indeed a good guide for future other evaluation steps. Yet nothing is said regarding future developments to be carried out in CLASSIC, in terms of improvement of already existing development or in terms of implementing new developments. Could you be more specific on this point ? Are there any weaknesses in CLASSIC that you suspect, or any characteristics known regarding the model responses to environmental conditions that this evaluation demonstrate they should be improved in the code ?

---

## Author Comment (AC2) · 11 Mar 2021

**Anonymous Referee (2)**

Overall assessment:

**RC1:** The manuscript submitted by Christian Seiler and coworkers present the evaluation of the open-source community land-surface model CLASSIC v1.0. A wide range of variables related to energy, water and carbon cycle are compared against observation-based, either site- or global-gridded, data.

An extensive evaluation and huge work have been carried out and are presented here with a high degree of clarity. The manuscript is very well organized and written, and a large and useful selection of figures and tables support this work. This manuscript is of strong interest, not only as an overview of the strengths and weaknesses of one specific model, CLASSIC, but also as a guide for the long but necessary land-surface model evaluation exercise. I detail here minor corrections and feedbacks to be considered, and I warmly recommend this manuscript for publication in GMD.

Reply: Thank you for your positive evaluation of our manuscript and the very encouraging remarks!

Corrections and feedbacks

- **RC2:** In the section 2.1 presenting the model tools, CLASSIC is among others presented as a dynamic vegetation model. However, in this work and generally, it is not clear to me if the vegetation distribution is indeed calculated "dynamically"

by the model, depending in particular on temperature and CO2 conditions, or prescribed based on land map forcings. Could the authors clarify this point in the manuscript? For a generally understanding as well: is nitrogen cycle included in this model?.

Reply: CLASSIC's biogeochemical component (CTEM) is a dynamic vegetation model. The spatial distribution of plant functional types (PFTs) can either be prescribed or simulated as a function of environmental conditions under which PFTs compete for light and water. However, it must be noted that competition is not the only dynamical aspect of vegetation dynamics. Other processes that affect vegetation dynamics include ecophysiology (allocation, turnover, mortality), disturbance (e.g. fire), and establishment, all of which are represented in CTEM. While competition can be understood as the spatial aspect of vegetation dynamics, all other processes that drive land-atmospheric fluxes present the vertical component of vegetation dynamics. The simulations presented here focus on the vertical component of vegetation dynamics. The model runs presented in this manuscript are based on a prescribed distribution of PFTs, which reduces the number of potential causes for biases. We now state that the spatial distribution of PFTs is prescribed and that the nitrogen cycle is turned off.

In section 2.1, we now write that: "In this study, the spatial distribution of PFTs has been prescribed to reduce the number of possible causes for model biases.".

Furthermore, we note that: "For the purpose of this study, CLASSIC's recently added nitrogen cycle has been turned off (Asaadi and Arora, 2021).".

- **RC2:** Correction page 4, line 114: a "c" in missing in "The protcol consists of a spin up".

Reply: Thank you, we now replaced *protcol* with *protocol*.

- **RC2:** Correction page 17, line 493: replace "most" by "more" in "the positive biases are most evident in the NH extratropics rather than in the tropics".

  Reply: We have now replaced *most* with *more*.

- **RC2:** The authors underline both in the Abstract and in the Conclusion that "Our results will serve as a baseline for guiding and monitoring future CLASSIC development." Regarding the development monitoring, this manuscript is indeed a good guide for future other evaluation steps. Yet nothing is said regarding future developments to be carried out in CLASSIC, in terms of improvement of already existing development or in terms of implementing new developments. Could you be more specific on this point? Are there any weaknesses in CLASSIC that you suspect, or any characteristics known regarding the model responses to environmental conditions that this evaluation demonstrate they should be improved in the code?

  Reply: We now list the main issues that future model development needs to address in the conclusion section: "The main deficiencies that should be addressed in future model development are the (i) positive albedo bias and resulting SW radiation bias in parts of the NH extratropics and Tibetan plateau, (ii) out-of-phase seasonal GPP cycle in the humid tropics of South America and Africa, (iii) lacking spatial correlation of annual mean NEE measured by FLUXNET sites, (iv) underestimation of fractional area burnt and corresponding emissions in the boreal forests, (v) negative soil organic carbon bias in high latitudes, and (vi) time lag in seasonal LAI maxima in the NH extratropics. Recent model development has started addressing some of those issues already, including the improvement of LAI seasonality through the incorporation of non-structural carbohydrates, which will form part of the next model version release (Asaadi et al. 2018). Further research is required to separate the impact of observational uncertainties on biases in LAI and above-ground biomass. For LAI, we propose to add additional observation-based reference data sets and reduce the spatial coverage to highquality grid cells that have not been gap filled. For biomass, we propose to extend our current forest inventory database to achieve greater spatial coverage.".

We went back to the results section to ensure that all items listed above are mentioned in their respective subsections. We also inserted a similar paragraph in the abstract.